# SLITRK1-mediated noradrenergic projection suppression in the neonatal prefrontal cortex

Minoru Hatayama [1,2], Kei-ichi Katayama[2], Yukie Kawahara[3], Hayato Matsunaga[1], Noriko Takashima[2], Yoshimi Iwayama[4], Yoshifumi Matsumoto[2], Akinori Nishi [3], Takeo Yoshikawa [4] & Jun Aruga [1,2✉]

*SLITRK1* is an obsessive-compulsive disorder spectrum-disorders-associated gene that encodes a neuronal transmembrane protein. Here we show that SLITRK1 suppresses noradrenergic projections in the neonatal prefrontal cortex, and SLITRK1 functions are impaired by SLITRK1 mutations in patients with schizophrenia (S330A, a revertant of *Homo sapiens*-specific residue) and bipolar disorder (A444S). Slitrk1-KO newborns exhibit abnormal vocalizations, and their prefrontal cortices show excessive noradrenergic neurites and reduced Semaphorin3A expression, which suppresses noradrenergic neurite outgrowth in vitro. Slitrk1 can bind Dynamin1 and L1 family proteins (Neurofascin and L1CAM), as well as suppress Semaphorin3A-induced endocytosis. Neurofascin-binding kinetics is altered in S330A and A444S mutations. Consistent with the increased obsessive-compulsive disorder prevalence in males in childhood, the prefrontal cortex of male Slitrk1-KO newborns show increased noradrenaline levels, and serotonergic varicosity size. This study further elucidates the role of noradrenaline in controlling the development of the obsessive-compulsive dis-order-related neural circuit.

[1] Department of Medical Pharmacology, Nagasaki University Institute of Biomedical Sciences, Nagasaki 852-8523, Japan. [2] Laboratory for Behavioral and Developmental Disorders, RIKEN Brain Science Institute, Wako-shi, Saitama 351-0198, Japan. [3] Department of Pharmacology, Kurume University School of Medicine, Kurume-shi, Fukuoka 830-0011, Japan. [4] Laboratory for Molecular Psychiatry, RIKEN Brain Science Institute, Wako-shi, Saitama 351-0198, Japan. ✉email: aruga@nagasaki-u.ac.jp

Recent genetic studies have reported an association of *SLITRK1* with Tourette's syndrome (TS)[1–4], trichotillomania (TTM)[5], and obsessive-compulsive disorder (OCD)[6,7]. Family and treatment studies have indicated that these three disorders comprise a larger spectrum of conditions (OCD spectrum disorder or obsessive-compulsive and related disorders, hereafter OCRD)[8–10]. Regarding OCD, recent studies have proposed specific neural circuits that could mediate cognitive and affective processing defects in patients with OCD[10]. In global terms, cortico-striato-thalamo-cortical (CSTC) circuits could be involved in OCD[10]. CSTC circuits comprise parallel and partly segregated circuits that are involved in sensorimotor, cognitive, affective, and motivational processes. Specifically, the CSTC circuits include the dorsomedial prefrontal cortex (PFC) and ventromedial PFC, which are partly related to the mouse medial PFC.

Monoaminergic neurotransmission in the brain is critical as a therapeutic target for OCRD. Selective serotonin reuptake inhibitor and clomipramine (serotonin–noradrenaline [NA] reuptake inhibitor) are used as first-line drugs for treating OCD[10]. Psychopharmacological studies have reported a more robust response of TTM and OCD to clomipramine than to desipramine (an NA reuptake inhibitor with weak serotonin reuptake inhibition, α1-blocking, antihistamine, and anticholinergic effects)[8]. Moreover, TS responds to haloperidol (a dopamine D2 receptor antagonist) and clonidine (an adrenergic α2 receptor agonist)[11].

Besides its importance as a therapeutic target, monoaminergic neurotransmission is involved in the etiology of OCRD. Some variants in serotonergic and catecholaminergic genes are associated with OCD and binding to serotonergic transporters or dopamine receptors was altered in imaging studies for OCD patients[10,12]. In rats, administration of clomipramine in neonates induces OCD-like behaviors in adulthood[13]. However, whether any OCRD-associated genes control the development of monoaminergic fibers remains to be clarified. There is accumulating evidence regarding the molecular mechanisms underlying dopaminergic and serotonergic fiber development in the developing brain. Numerous axon guidance molecules, including Wnts, ephrins, Slit-Robo, netrins, semaphorins, protocadherins, and neurotrophins, control the development of monoaminergic projections during rodent embryonic development. Contrarily, the mechanism of developmental control underlying NA projections originating from diverse neurons in the hindbrain remains unclear[14].

SLITRK1 is a leucine-rich repeat (LRR)-containing single-path transmembrane protein that is predominantly expressed in the mammalian brain. Moreover, it is a SLITRK family member, which is comprised of six members (SLITRK1–6)[15,16]. Studies have demonstrated molecular functions associated with the control of neurite development and synaptogenesis[1,15,17–22]. The binding of a signaling adaptor protein 14-3-3β (YWHAB) to the cytoplasmic C-terminal domain is involved in neurite outgrowth[17]. Further, *trans*-interactions between receptor-type protein tyrosine phosphatases and the distal LRR domain (LRR1) have been proposed to mediate synaptogenic activities[19,20,23]. Additionally, SLITRK1 can be cleaved by α/γ secretase at the transmembrane region, with the cleaved extracellular domain (ECD) being secreted[17]. However, the physiological significance of the cleaved SLITRK1 ECD remains unclear.

We previously reported that adult Slitrk1 knockout (KO) mice present with anxiety and depression-like behavior, as well as increased NA levels in the prefrontal cortex (PFC)[24]. Furthermore, clonidine attenuated anxiety-like behavior in Slitrk1-deficient mice. This suggested the involvement of noradrenergic mechanisms in the behavioral abnormalities of Slitrk1-KO mice. The efficacy of clonidine confirmed the predicted validity of Slitrk1-KO mice as an animal model of TS. However, regarding face (symptomatic) validity, there were no OCRD-like behavioral abnormalities in Slitrk1-deficient mice[24]. Major depressive disorder and anxiety disorders are major comorbidities in TS[25] and OCD[26,27]. The inconsistency in symptoms between Slitrk1-KO mice and patients with OCRD suggested latent OCRD-related abnormalities in the animal model. No study has described any Slitrk1-KO-related neurodevelopmental phenotypes. Furthermore, SLITRK1 could be involved in the pathogenesis of other neuropsychiatric disorders involving depression and anxiety.

In this study, we firstly examined the neonatal phenotypes of Slitrk1-deficient mice. Because abnormalities in NA fiber development were observed, we investigated the molecular mechanism underlying Slitrk1-mediated control of neurite development. Further, we conducted a re-sequencing analysis of patients with schizophrenia (SCZ) and bipolar disorder (BPD) to identify functionally defective and significantly enriched missense mutations. The analysis identified a SLITRK1 mutation that affects the NA fiber development-controlling and L1 family protein-binding abilities of SLITRK1. Finally, we sought to discuss the pathogenesis of OCRD, focusing on the role of neonatal NA-mediated neural circuit modification.

## Results

### Abnormalities in the developing Slitrk1-KO mice

To elucidate the pathophysiology of SLITRK1-associated disorders, we examined the neurodevelopmental phenotype of the Slitrk1-KO mice. First, we observed altered body weight in developing Slitrk1-KO mice (Fig. 1a). Male Slitrk1-KO mice aged 8 weeks (W) have been shown to maintain 11% lower body weight compared with wild-type (WT) mice[24]. During development, KO male mice showed lower body weights as early as P3 ($-12\%$, $P = 0.022$; WT, $n = 22$; KO, $n = 24$), which disappeared at P7 and P14, and returned from P21 ($-14\%$, $P < 0.01$) to later stages with a 9–14% lower body weight than that of WT mice at the same stage. Compared with WT mice, female KO mice showed a lower body weight only at P14 ($-17\%$, $P < 0.01$) during postnatal development. The sex-dependent body weight differences suggested that Slitrk1 is involved in a sex-specific development-controlling mechanism.

Behavioral tests on developing rodents revealed no clear abnormalities in reflexes (righting, pivoting, rooting, geotaxis, bar holding, grasping, visual place response, auditory startle, and tactile startle) considered as developmental milestones[28,29] (Supplementary Figs. 1 and 2).

Regarding locomotion, there were no clear differences until 4W, with the exception of 5W and 6W, where KO mice showed lower locomotion activity than WT mice (Fig. 1b). Since adult Slitrk1-KO mice have decreased locomotor activity[24], we hypothesized that the adult Slitrk1-KO-like behavior would manifest at the 5W stage. Consistent with this hypothesis, KO mice at 5W presented with a longer immobility time than WT mice in the forced-swimming test (Fig. 1c), which is consistent with previous findings in adult Slitrk1-KO mice[24].

Further, we examined isolation-induced ultrasonic vocalization (USV) calls of the infant mice. In early mouse development, there is an increase in the call rate (number/min) for the first 5 days, which peaks between 5 days old (P5) and P11[30]. The call rate of female KO mice was significantly lower than that of WT mice at P7 ($P = 0.0029$) and P10 ($P = 0.0012$) (Fig. 1d). The maximal vocalization amplitude of KO mice was lower than that of WT at P4 (male, $P = 0.035$; mixed-sex, $P = 0.023$) and P7 (male, $P = 0.0032$; female, $P = 0.0043$) (Fig. 1d and Supplementary Fig. 3). For the sound frequency analysis of the vocalizations, we categorized USV syllables into five groups (upward, downward, flat, inverted U-shaped, and U-shaped); further, there was a significant decrease in the inverted U-shaped syllable in KO mice

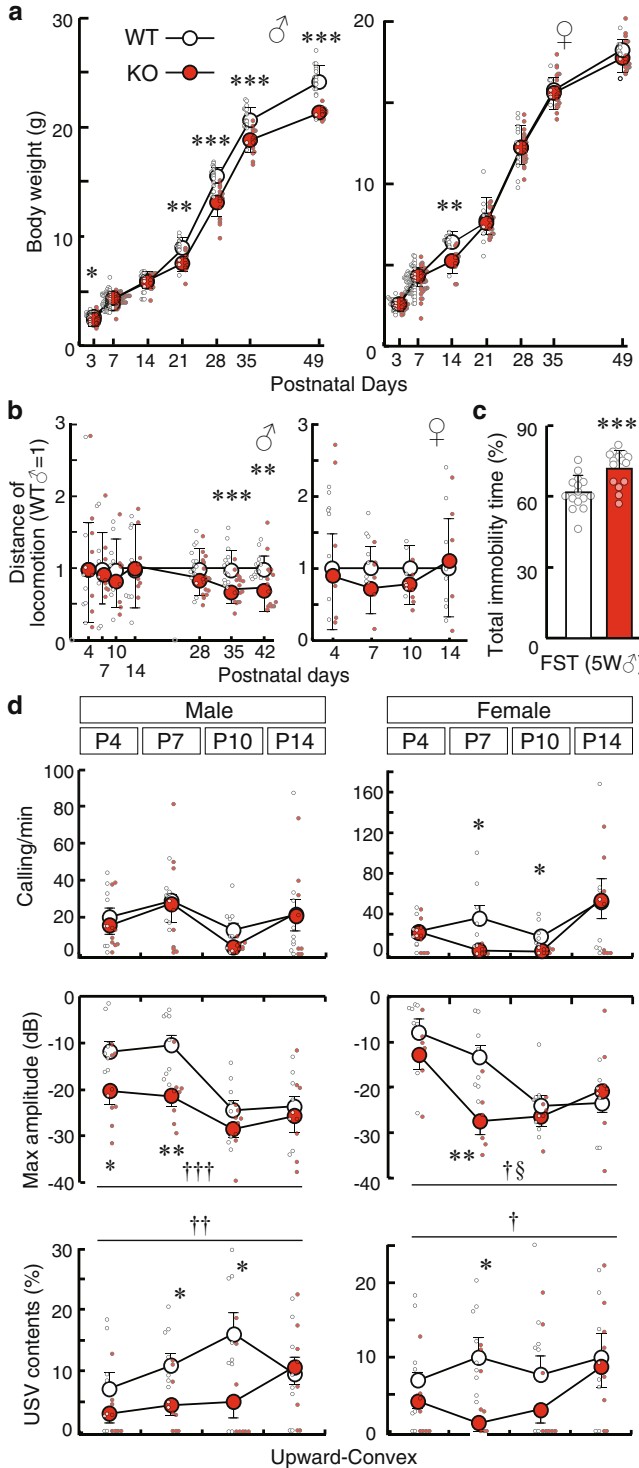

**Fig. 1 Early postnatal development and behavior of Slitrk1-KO mice.**
**a** Body weight change during postnatal development (P3–P49). ♂, male; ♀, female; open circle, WT; red circle, KO. WT male, n = 17, 62, 19, 15, 20, 19, 19; KO male, n = 21, 53, 14, 13, 20, 17, 17, 17; WT female, n = 20, 70, 12, 9, 13, 8, 12, 9, 8, 13, 8, 8, 8; KO female, n = 22, 42, 12, 24, 26, 23, 23, 23 at P3, P7, P14, P28, P35, P49 respectively. **b** Locomotor activities: The total moving distance in 3 min (P4–P14) or 10 min (P28–P42) are indicated as relative values, where the WT mean at each time point equals to 1. ♂, male; ♀, female; open circle, WT; red circle, KO. WT male, n = 8 (P4–P14), n = 15 (P28, P35), n = 14 (P49); KO male, n = 8 (P4–P14), n = 18 (P28–P42), WT female, n = 8 (P4–P14); KO female, n = 6 (P4–P10), n = 5 (P14). **c** Immobility time in the forced swimming test. WT, n = 15; KO, n = 18, 5 weeks-old male mice. Open bar, WT; red bar, KO. **d** USV of isolated pups. Open circle, WT; red circle, KO. WT male, n = 9; KO male, n = 8; WT female, n = 8; KO female, n = 6. Calling rate, maximum amplitude, and percentage of the upward-convex type vocalizations among total vocalizations are indicated. Data were presented as mean ± standard deviation (SD). *P < 0.05; **P < 0.01; ***P < 0.001 in t-test. †P < 0.05; ††P < 0.01; †††P < 0.001 in two-way ANOVA (genotype and day as main factors, genotype effect). §P < 0.05 in two-way ANOVA (genotype × day interaction).

at P7 (male, P = 0.027; female, P = 0.013) and P10 (male, P = 0.026) (Fig. 1d and Supplementary Fig. 3).

These findings revealed phenotypes of Slitrk1-KO mice during postnatal development. Compared with WT mice, KO mice showed lower body weight (P3, males) and vocalization abnormalities (P4–P10). The lower locomotion and depression-like behavior in adult Slitrk1-KO-like manifested as early as 5W of age. Given the role of monoaminergic disturbances in the observed phenotypes (see the "Discussion" section), we subsequently investigated the monoaminergic nervous system in developing Slitrk1-KO mice.

**Slitrk1-KO mice present with altered monoaminergic fiber morphology.** There were no obvious abnormalities in the histological architecture of the brains of Slitrk1-KO neonatal mice. A previous study reported that adult male Slitrk1-KO mice showed increased levels of NA and its metabolite, 3-methoxy-4-hydroxyphenylglycol (MHPG), in the PFC and nucleus accumbens (NAC), as well as increased levels of 5HT metabolite, 5-hydroxyindoleacetic acid (5-HIAA), in the NAC[24]. Furthermore, monoaminergic drugs or receptor deficiency affect infant body weight and USV[31–33]. Therefore, we examined the distribution of molecular markers related to monoaminergic projections in the PFC and NAC using antibodies against the NA transporter (NET), serotonin transporter (SERT), and dopamine transporter (DAT). Additionally, we stained cholinergic fibers using an anti-choline acetyltransferase (ChAT) antibody, with the analyses focusing on the P7, 4–5W, and adult (6M) stages.

There were morphological alterations of noradrenergic projections at the neonatal stage. In the superficial layer of the PFC, there was an increased density of tangential NET-positive fibers during neonatal development (P0–P5) (Supplementary Fig. 4). At P7, the PFC of Slitrk1-KO mice showed a two-fold increased density of the NET-positive fibers compared with that in WT mice in each sex (male, P = 0.0040; female, P = 0.045) (Fig. 2a, b). The corresponding WT-KO difference was not clear at P3, 5W, or 6M (Fig. 2a), indicating enhanced NET-positive fiber growth temporarily between P3 and P7 in the PFC of Slitrk1-KO mice. However, the varicosity size of NET fibers showed different developmental profiles; specifically, it was 9.2% larger in KO male mice at P7 (P = 0.044, Supplementary Fig. 5) and 7.7% smaller at 6M (P = 0.0086) compared with the size in WT controls (Fig. 2c). Consistent with the increased NET-positive fiber density at P7, there was an increased size of the NET-positive locus coeruleus (LC) area in KO mice at P7 (male, +36%, P = 0.036; female, +17%, P = 0.10) (Fig. 2d, e).

The varicosity size of SERT-positive fibers was 21% larger in the PFC of male KO mice at P7 (P = 0.0087) than that of WT mice, but not in female at P7 or in male or female at 6M (Fig. 3a–c). Additionally, the SERT-positive varicosity size tended to be larger in the NAC of KO mice at P7 (P = 0.086, Fig. 3d–g) than that in WT mice. There were no significant between-group differences in the varicosity size of DAT-positive dopaminergic fiber in the NAC at P7 (Fig. 3h–j). ChAT-positive cholinergic

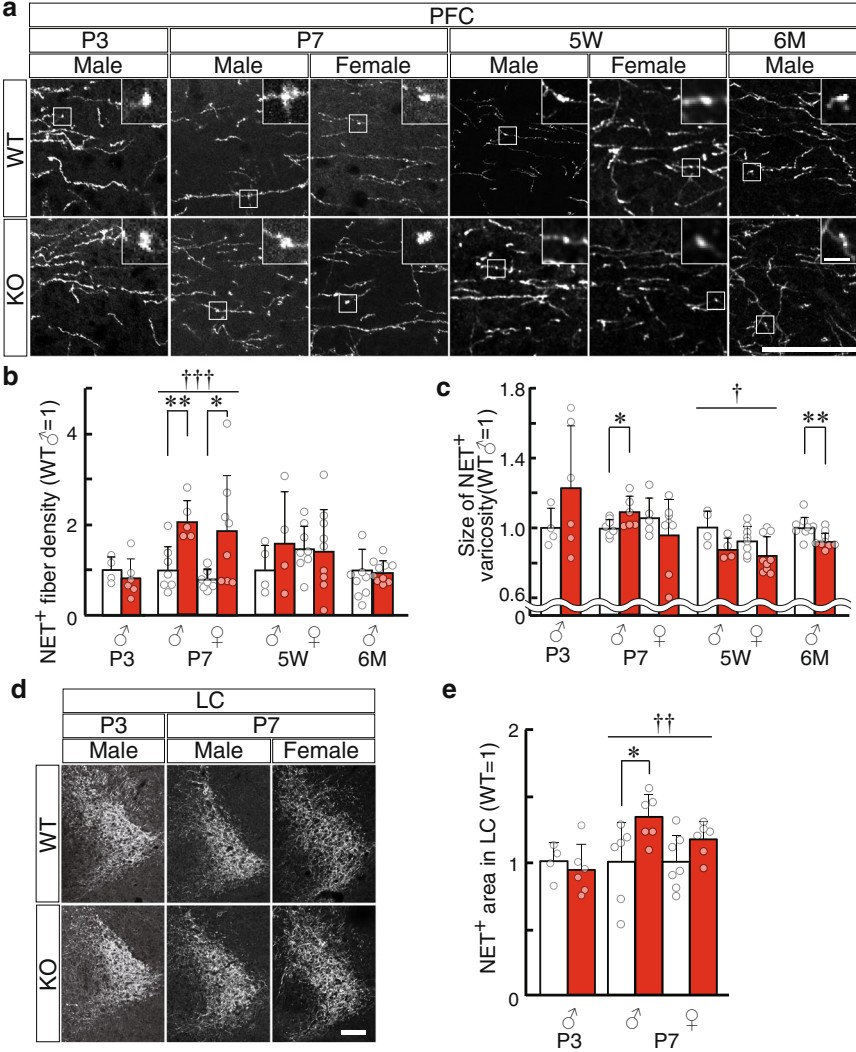

**Fig. 2 Postnatal noradrenergic fiber development and monoamine levels in the brains of Slitrk1-KO mice.** Brain sections obtained from 3 days-old (P3), 7 days-old (P7), 5 weeks-old (5W), or 6 months old (6M) mice were immunostained using an anti-NET antibody. **a** Representative image for PFC. Insets indicate the enlarged views of the varicosities in the boxed region. *Scale bar*, 50 μm; 5 μm (inset). **b** NET$^+$ fiber density in the PFC. The mean value of male WT mice was defined as 1. **c** Size of NET$^+$ varicosity. The mean value of male WT mice was defined as 1. Absolute values of the male WT varicosity size in μm2: P3, 0.90; P7, 077; 5W, 1.33; 6M, 1.02. **b**, **c** WT male, $n = 4, 7, 4, 9$; KO male, $n = 6, 5, 4, 9$ at P3, P7, 5W, 6M, respectively; WT female, $n = 8, 8$; KO female, $n = 8, 9$ at P7, 5 W respectively. **d** Representative images for LC. *Scale bar*, 100 μm. **e** NET$^+$ area in the LC. The mean value of WT mice was defined as 1. Absolute values of the male WT varicosity size in μm2: P3, $1.3 \times 10^4$; P7 male, $8.9 \times 10^3$; P7 female $= 1.2 \times 10^4$ μm2. WT male, $n = 4, 6$; KO male, $n = 6, 6$ at P3, P7, respectively. WT female, $n = 7$; KO female, $n = 6$ at P7. **b**, **c**, **e** *Open bar*, WT; *red bar*, KO. Data were presented as mean ± SD. *$P < 0.05$; **$P < 0.01$ in *t*-test. †$P < 0.05$; ††$P < 0.01$; †††$P < 0.001$ in two-way ANOVA (genotype and sex as main factors, genotype effect).

fiber showed an increase in the varicosity size in the adult PFC (Fig. 3k, l).

Given the morphological changes of monoaminergic fibers in the brains of KO mice at P7, we measured the PFC and striatal levels of the monoamine transmitters and their metabolites at P7 (Fig. 4a–d, Supplementary Table 1). In the PFC, there was an 89% increase in NA levels in males ($P = 0.0063$), but not in females ($+12\%$, $P = 0.62$), KO mice ($F(1,22) = 1.96$, $P_{genotype \times sex} = 0.18$ in genotype × sex two-way analysis of variance [ANOVA]) (Fig. 4c). Both sexes of KO mice showed decreased levels of NA metabolite (MHPG) and NA turnover (MHPG/NA) (MHPG, $-81\%$, $F(1,21) = 6.12$, $P_{genotype} = 0.022$; MHPG/NA, $-78\%$, $F(1,21) = 5.64$, $P_{genotype} = 0.027$ in genotype × sex two-way ANOVA) (Fig. 4c, d). There was no between-group difference in dopamine levels; however, male KO mice showed increased levels of its metabolite, homovanillic acid ($+75\%$, $P = 0.017$, Fig. 4c). Increased NA levels could have affected dopamine

metabolite levels since the same set of enzymes catabolize NA and dopamine. There were no significant between-group differences in monoamine transmitters or their metabolites in the striatum at P7 (Supplementary Table 1).

The above results revealed various molecular markers or neurochemical phenotypes in the PFC of Slitrk1-KO neonatal mice. However, the among-phenotype causal relationship remains unclear. Given the sex specificity of the observed phenotypes and the sexual dimorphism of monoamine metabolism (see the "Discussion" section), we hypothesized that unique sex-independent increases in noradrenergic fibers may have caused other phenotypes. To test this, we administered clomipramine (15 mg/kg/day), which is a 5HT and NA reuptake inhibitor, or clonidine (0.1 mg/kg/day), which is an α2 adrenergic agonist, during P3 to P6 and examined the effects at P7 (Fig. 5a–c). There was increased serotonergic varicosity size (male, $+2.6\%$, $P = 0.32$; female, $+4.1\%$, $P = 0.0098$; mixed-sex,

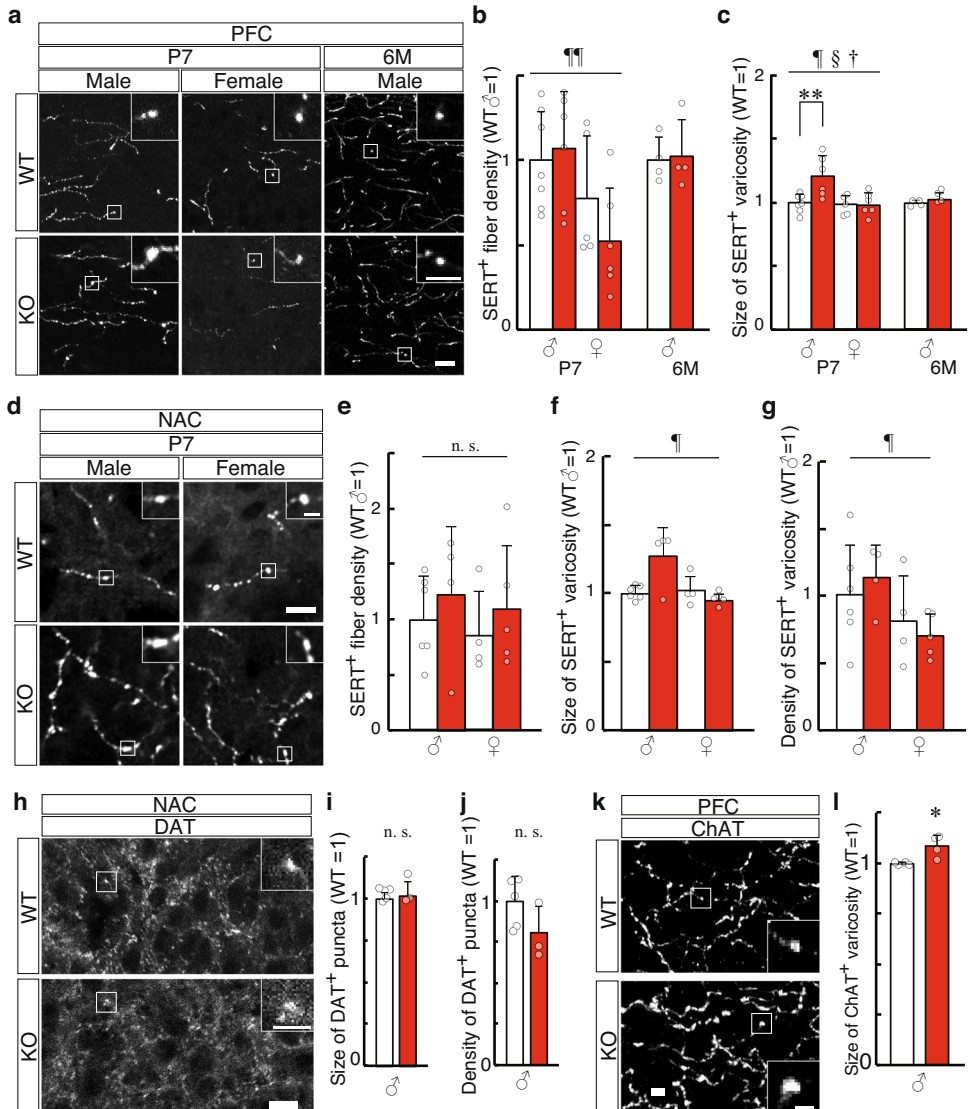

**Fig. 3 Phenotypes of serotonergic, dopaminergic, and cholinergic fibers in developing Slitrk1-KO mice. a–g** Serotonergic fiber in the PFC (**a–c**) and NAC (**d–g**) were analyzed by anti-SERT immunostaining. **a** Images of the PFC in 7 days-old (P7) or 6 months-old (6M) WT (P7 male, $n = 7$; P7 female, $n = 5$; 6M male, $n = 4$) and KO (P7 male, $n = 6$; P7 female, $n = 6$; 6M male, $n = 4$) mice. Insets indicate the enlarged views of the varicosities in the boxed region. Density (**b**) and varicosity size (**c**) of SERT⁺ fibers were measured. **d** Images of the NAC in WT (male, $n = 6$; female, $n = 4$) and KO ($n = 4$, male; $n = 5$, female) mice at P7. Density (**e**), varicosity size (**f**), and varicosity density (**g**) of SERT⁺ fibers were analyzed. **h–j** Dopaminergic fibers in the NAC were analyzed at P7 by anti-DAT immunostaining. **h** Images of the NAC in WT Size (**i**) and density (**j**) of DAT⁺ puncta were analyzed. WT, $n = 5$; KO, $n = 3$. **k, l** Cholinergic fibers in the PFC were analyzed at 6M with anti-ChAT immunostaining. **k** Images of the PFC. **l** Size of ChAT⁺ varicosities. *$P < 0.05$ in t-test, $n = 4$ per genotype. (*All graphs*) Open bar, WT; red bar, KO. The average of male WT mice was defined as 1 at each stage. Absolute values of the male WT varicosity size in μm²; **c** (SERT, PFC) P7, 0.79; 6M, 0.60; **f** (SERT, NAC) 0.70; **i** (DAT) 0.48, **l** (ChAT) 0.60. Data were presented as mean ± SD. *$P < 0.05$; **$P < 0.01$ in t-test. †$P < 0.05$ (genotype); ¶$P < 0.05$; ¶¶$P < 0.01$ (sex); §$P < 0.05$ (genotype × sex interaction) in two-way ANOVA (genotype and sex as main factors). *Scale bars*, 10; 5 μm (insets in **a, h, k**); 2 μm (inset in **d**).

+3.3%, $P = 0.028$) (Fig. 5a, c). The results suggested that noradrenergic fiber overgrowth may have partly caused the other sex-dependent phenotype.

**Semaphorin3A (Sema3A) expression was reduced in the PFC of Slitrk1-KO neonatal mice.** To elucidate how Slitrk1 deficiency causes noradrenergic projection overgrowth, we conducted a quantitative polymerase chain reaction (PCR) analysis for known axon guidance molecules for monoaminergic fibers. There were significantly decreased Sema3A, Slit2, and Slit3 mRNA levels in Slitrk1-KO mice (Sema3a, −11%, $P = 0.014$; Slit2, −15%, $P = 0.017$; Slit3, −14%, $P = 0.025$; WT, $n = 13$; KO, $n = 13$) (Fig. 6a and Supplementary Table 1). Subsequently, we examined

the Sema3A-Fc effects on the neurite outgrowth of cultured LC neurons from WT mice. Sema3A-Fc suppressed the neurite complexity and total branch length (Fig. 6b, c-*WT*, d) compared with the carrier protein (bovine serum albumin [BSA])-only control. We also investigated the effect of SLITRK1 ECD because SLITRK1 ECD is known to be cleaved by α/γ secretase at the transmembrane region[17]. The same assay revealed that SLITRK1 ECD protein exerted considerable neurite suppressive activity (Fig. 6b, c-*WT*, d). These findings suggest that both Sema3A-Fc and Slitrk1 ECD can suppress noradrenergic neurites development.

Further, we performed the same assay using LC neurons from Slitrk1-KO mice. Control treatment led to no change in the neurite length of KO LC neurons (Fig. 6b, d). However, compared

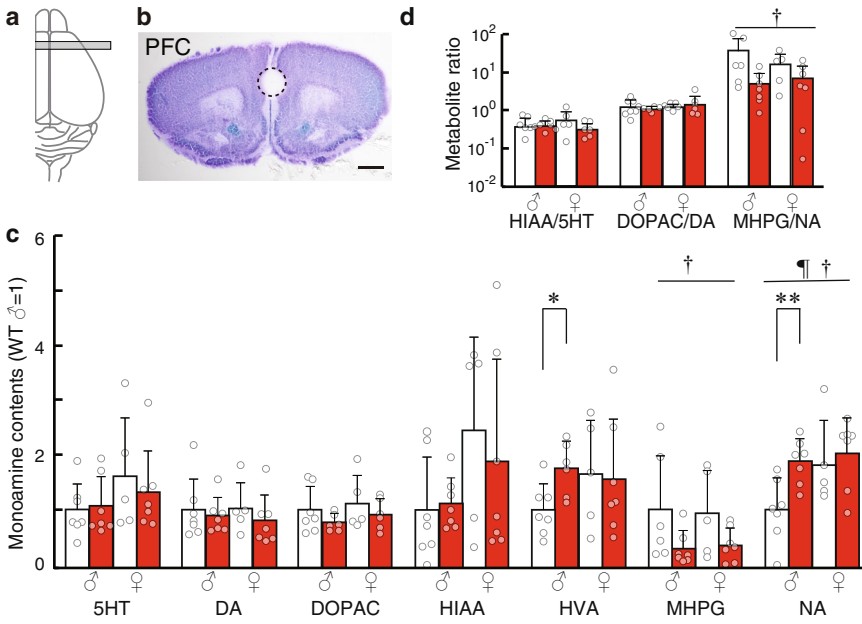

**Fig. 4 Monoamine levels measured through high-performance liquid chromatography. a** Coronal brain sections were prepared from the illustrated areas. **b** Cresyl violet staining of the sections used for sampling. Positions of the punch biopsy are indicated by circle. **c** Monoamine contents in P7 WT and KO mice. The mean value of male WT mice was defined as 1. Absolute values are indicated in Supplementary Fig. 5. 5HT serotonin; DA dopamine, DOPAC 3,4-dihydroxyphenylacetic acid, HIAA 5-hydroxyindole acetic acid, HVA 3-methoxy-4- hydroxyphenyl acetic acid, MHPG 3-methoxy-4-hydroxyphenylglycol, NA noradrenaline. **d** Monoamine turnover ratios. The ratios mean (metabolite content)/(monoamine content). $n = 5$–7 mice per genotype. Outliers were removed after Grubbs's test. Data were presented as mean ± SD. *$P < 0.05$; **$P < 0.01$ in $t$-test. $†P < 0.05$; $††P < 0.01$; $†††P < 0.001$ in two-way ANOVA (genotype and sex as main factors, genotype effect). $¶P < 0.05$ in two-way ANOVA (genotype and sex as main factors, sex effect). *Scale bar*, 100 µm.

with WT LC neurons, the KO LC neurons showed higher and lower neurite complexity in the proximal and distal neurites, respectively (Fig. 6b, e-*BSA*). Compared with the control treatment, Sema3A-Fc treatment led to decreased and increased complexity of proximal and distal neurites, respectively (Fig. 6b, c-*KO*). However, there was no significant between-genotype difference in the complexity (Fig. 6b, e-*Sema3a-Fc*). There was a negligible difference between SLITRK1 ECD and control treatments (Fig. 6b, c-*KO*, d). Further, compared with WT mice, SLITRK1 ECD treatment of KO neurons showed increased complexity of proximal neurites in LC neurons (Fig. 6b, e-*SLITRK1 ECD*). Taken together, our findings suggested that Slitrk1 has both cell-autonomous and cell-nonautonomous functions for controlling noradrenergic neurite development and that Slitrk1 in the LC is necessary for a proper response to the Sema3a or SLITRK1 ECD.

**Identification of SLITRK1 missense mutations in patients with SCZ and BPD.** Based on the behavioral abnormalities in adult Slitrk1 KO mice (see the "Introduction" section), we examined the possible involvement of SLITRK1 in neuropsychiatric disorders other than OCRD. Accordingly, we conducted a re-sequencing analysis on the SLITRK1-coding region and its flanking region of genomic DNA obtained from Japanese patients with SCZ ($n = 1040$) and BPD ($n = 364$), as well as healthy controls ($n = 1047$). There were four missense mutations (S330A, D348Y, G352R, A444S) in patients with SCZ and BPD (Fig. 7a, b, Supplementary Table 2). S330A and A444S were distributed with high frequency in the SCZ/BPD and BPD cohorts, respectively. The minor allele frequency (MAF) values for S330A were 0.013, 0.015, and 0.0049 in the SCZ, BPD, and control cohorts, respectively. A444S was detected heterogeneously in 2 out of 364 patients with BPD (MAF = 0.0027) but not in the control or SCZ cohorts (MAF = 0). In the current human genetic variant database (http://

www.ensembl.org/Homo_sapiens/Gene/Variation_Gene/), which includes the findings of the 1000 genome project[34], there were no known mutations for A444; however, S330A was listed using a global MAF value of 0.002.

Phylogenetic analysis (Fig. 7a) revealed that A444 was conserved among the eutherians (placental mammals; human, dog, and mouse); moreover, threonine was at this position (T444) in the other vertebrates. Contrastingly, S330 was only detected in humans; moreover, alanine (A330) was at the same position in other primates, including *Homo neanderthalensis*. Therefore, the S330A mutation can be described as a reversion of the modern human-specific evolutionary trait. Regarding the domains in the SLITRK1 protein, S330 was located on the linker region between the conserved distal (LRR1) and proximal (LRR2) LRR domains; moreover, A444 was mapped in the LRR2 domain (Fig. 7a).

**The neurite-controlling function was affected by patient-derived SLITRK1 mutations.** Subsequently, we examined the effects of S330A and A444S on the SLITRK1 function. Expression constructs for human SLITRK1 WT, S330A, and A444S were prepared together with known variants L422fs[1], R584K, and S593G[5]. All constructs were designed to express the full-length protein using an endogenous signal sequence. Both the intact and N-terminally HA-tagged versions were expressed.

First, we examined the subcellular localization of S330A and A444S. There were between-group differences in the distribution in hippocampal neurons or COS7 cells (Supplementary Fig. 6). Additionally, the protein stability of S330A and A444S was comparable to that of SLITRK1 WT in PC12 cells, with no difference in excised ECD levels between the variants and WT (Supplementary Fig. 6). There were no clear differences in endoplasmic reticulum (ER) stress, which is often caused by misfolded protein (Supplementary Fig. 6). Our findings indicated

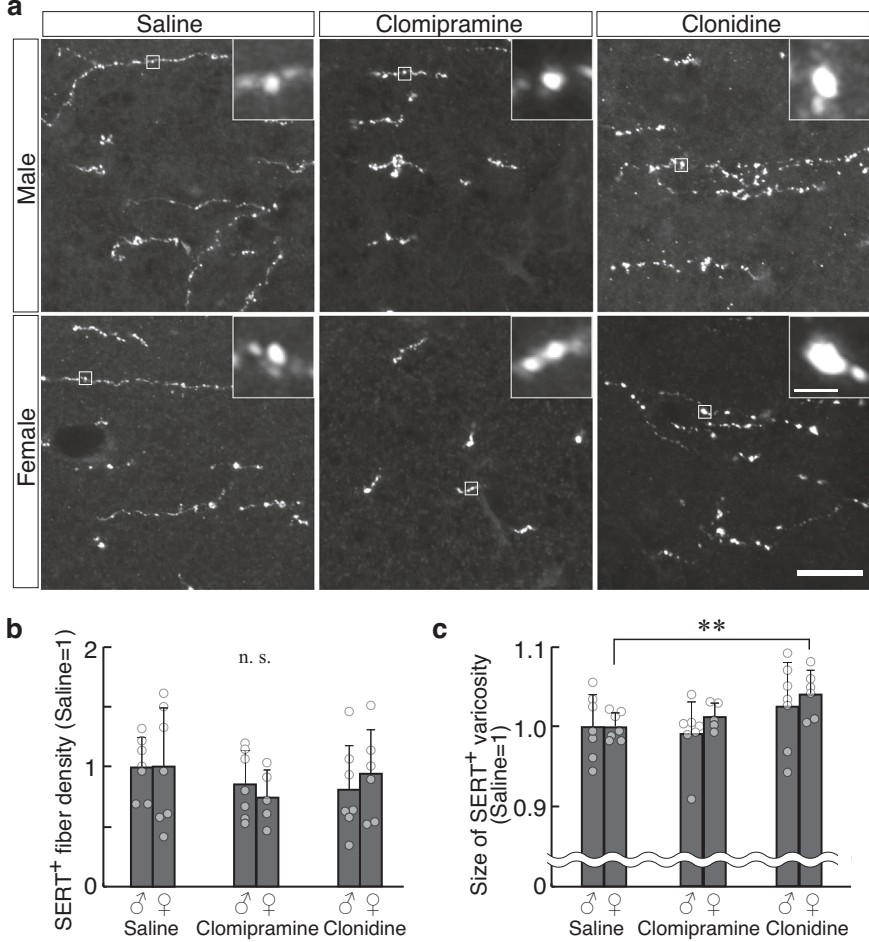

**Fig. 5 Effects of clomipramine or clonidine treatment on infantile serotonergic fiber morphology. a** Representative images of the serotonergic fibers in the PFC of P7 mice treated with saline, clomipramine (15 mg/kg/day), and clonidine (0.1 mg/kg/day) during P3–P6. The fiber density (**b**) and varicosity size (**c**) of SERT$^+$ fibers were compared with those of saline-treated samples. Male, $n = 7, 7, 7$; female, $n = 7, 5, 6$ (saline, clomipramine, clonidine, respectively). In both analyses, the mean value for the saline-treated mice was defined as 1. *Scale bar*, 10; 1 μm (inset). Data were presented as mean ± SD. **$P < 0.01$ in Dunnett's test.

that neither S330A nor A444S causes protein production, trafficking, or stability deficit.

Next, we examined the S330A and A444S SLITRK1 protein functions. Regarding neurite outgrowth-controlling activities, we conducted two assays using hippocampal and cortical neurons (Fig. 8a)[1]. For the cortical neuron assay (Fig. 8a–d), expression plasmids were introduced into the cerebral cortex at embryonic day (E) 14.5 using in utero electroporation. Subsequently, the transfected cortical neurons were cultured at E15.5 and the neurite pattern of the cortical neurons was examined at 4 days in vitro (DIV 4). In the hippocampal neuron assay (Fig. 8a, e–g), expression plasmids were transfected into neurons at DIV 8, followed by an examination at DIV 10. The cortical neuron assay, rather than the hippocampal neuron assay, can assess the overexpression effect on earlier events of neurite development (Fig. 8a). In both assays, SLITRK1 WT suppressed both neurite length and the number of early neurites. Compared to the WT activities, A444S exhibited a stronger suppression effect on the early neurite length and numbers (Fig. 8b–d); however, it exerted weaker suppression on the late neurite length and numbers (Fig. 8e–g). S330A exhibited weaker suppression on the later neurite length and complexity (Fig. 8e–g).

Overall, our findings suggest that the neurite-controlling activities of S330A and A444S differed from those of WT.

Specifically, the effects on late neurite development were affected by both mutations.

**Inhibitory synapse-inducing ability was lower in patient-derived mutants**. Another known function of SLITRK family proteins is synaptogenesis enhancement[18,23]. Accordingly, we quantified the VGAT- or VGLUT1-positive signals on the primary cultured hippocampal neurons co-cultured with HEK293T cells transfected with SLITRK1. Compared with an empty vector (GFP), SLITRK1 WT overexpression induced more VGAT-positive synapses; however, S330A induced fewer VGAT-positive synapses ($P = 0.019$) (Supplementary Fig. 7). Furthermore, A444S exhibited non-significantly lower synapse-inducing activity ($P = 0.057$) (Supplementary Fig. 7). In these experimental conditions, SLITRK1 and SLITRK2 WT exhibited negligible and significant, respectively, VGLUT1-positive excitatory synapse-inducing activity (Supplementary Fig. 7). These results suggested that S330A and A444S impair the inhibitory synapse-inducing ability of SLITRK1.

**SLITRK1 can alter noradrenergic fiber development**. The aforementioned findings suggested that SLITRK1 may be involved in controlling NET-positive neurite development. Subsequently, we

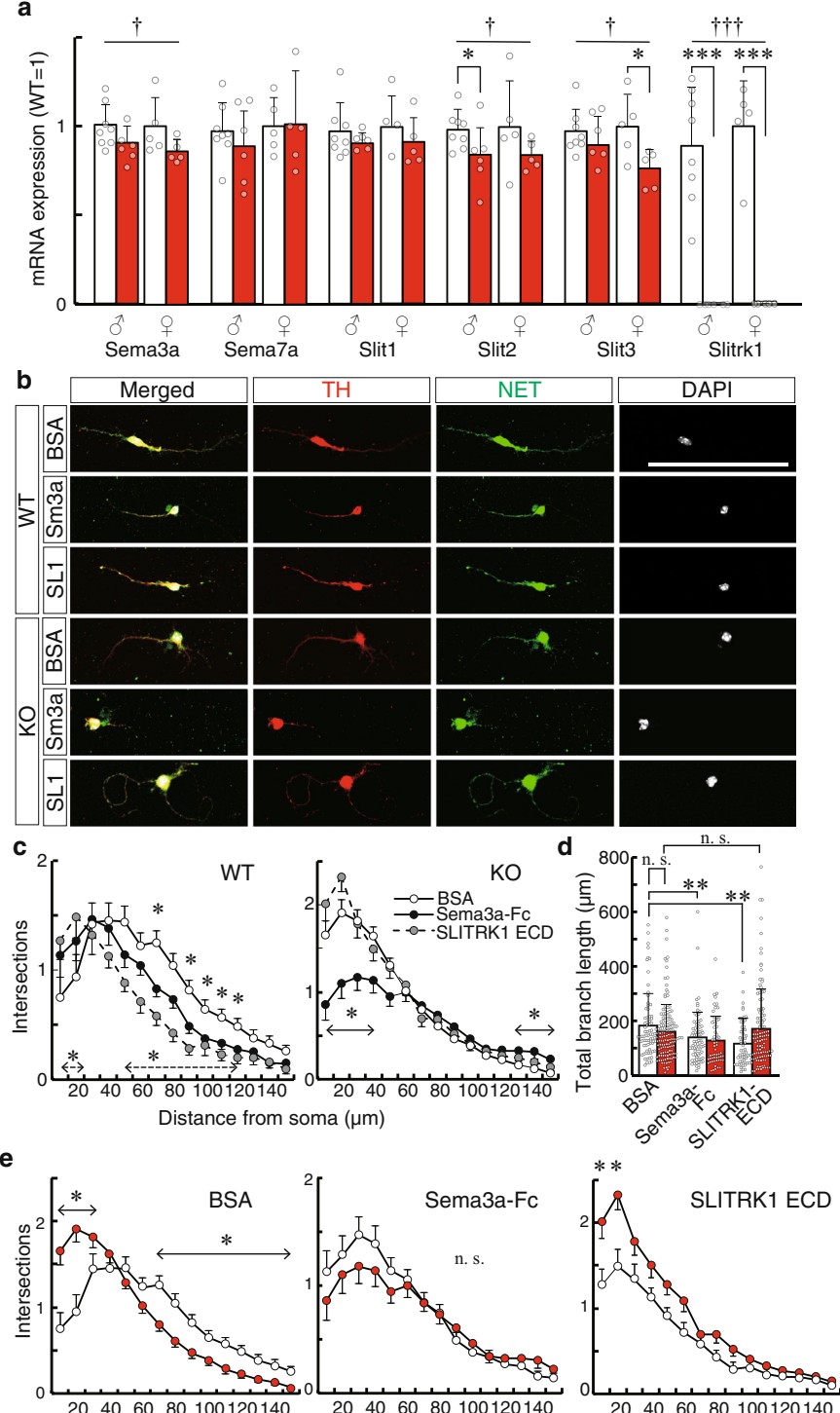

**Fig. 6 Sema3a and SLITRK1 ECD suppress neurite growth of LC neurons. a** Quantitative PCR of the P3 PFC. Expression levels were normalized using the WT mean values for each sex ($n = 8$ mice per male group, 5 mice per female group). Data were presented as mean ± SD. The actual values and those of P7 PFC are indicated in Supplementary Table 1. **b** Representative images of the primary culture of LC neurons with BSA (WT, $n = 74$ cells; KO, $n = 139$ cells), Sema3a (WT, $n = 77$ cells; KO, $n = 58$ cells), or SLITRK1 ECD (WT, $n = 62$ cells; KO, $n = 109$ cells). Neurons were stained using anti-Tyrosine hydroxylase (TH, red) and NET (green). Nuclear staining was performed using DAPI (gray). *Scale bar*, 100 µm. **c** Sholl analysis of neurites. Values are presented as mean ± SEM. *Open circle*, BSA treatment; *filled circle*, Sema3a treatment; *gray circle*, SLITRK1 ECD treatment. Statistical tests were performed in comparison to BSA at each distance (*line with double-headed arrow*, Sema3a; *broken line with double-headed arrow*), SLITRK1 ECD). **d** The total branch length. Data were presented as mean ± SD. *Open bar*, WT; *red bar*, KO. **e** Sholl analysis was reorganized to show the between-genotype differences. Values are presented as mean ± SEM. *Open circle*, WT; *red circle*, KO. *$P < 0.05$; **$P < 0.01$; ***$P < 0.001$ in $t$-test (**a**), Steel's test (**c**), Dunnett's test (**d**), or $U$-test (**e**). The between-genotype differences in Sema3a, Slit2, and Slit3 in (**a**) were significant after Benjamini−Hochberg correction for multiple comparisons. To consider multiple tests in (**a**),†$P < 0.05$; †††$P < 0.001$ (genotype) in two-way ANOVA (genotype and sex as main factors).

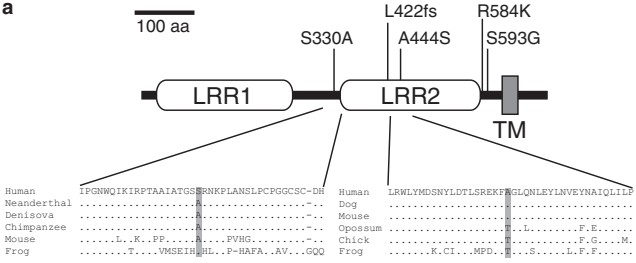

**Fig. 7 Screening of SLITRK1 missense mutations in patients with a neuropsychiatric disorder. a** The domain structure of human SLITRK1 and patient-derived missense mutations. S330A (BPD/SCZ, this study), A444S (BPD, this study), L422fs (Tourette's syndrome)[1], R584K/S593G (Trichotillomania)[5]. aa amino acid, LRR leucine-rich repeat, TM transmembrane domain. **b** Results of resequencing analysis of Japanese patients with SCZ and BPD. $P$ values indicate those obtained in $\chi^2$ test.

**Re-sequencing analysis of Japanese schizophrenia or bipolar patients**

| Polymorphism | | Genotype count | | | $P$ value | Allele count | | $P$ value |
|---|---|---|---|---|---|---|---|---|
| | | T/T | T/G | G/G | | T | G | |
| p.Ser330Ala | Schizophrenia | 1014 | 26 | 0 | **0.0307** | 2054 | 26 | **0.0315** |
| (c.988 T>G) | Bipolar | 350 | 11 | 0 | **0.0204** | 711 | 11 | **0.0210** |
| | Control | 1047 | 13 | 0 | | 2107 | 13 | |
| | | G/G | G/T | C/T | | G | T | |
| p.Asp348Tyr | Schizophrenia | 1040 | 0 | 0 | 1.0000 | 2080 | 0 | 1.0000 |
| (c.1,042 G>T) | Bipolar | 363 | 1 | 0 | 0.0878 | 727 | 1 | 0.0879 |
| | Control | 1060 | 0 | 0 | | 2120 | 0 | |
| | | G/G | G/C | C/C | | G | C | |
| p.Gly352Arg | Schizophrenia | 1038 | 2 | 0 | 0.1532 | 2078 | 2 | 0.1533 |
| (c.1,054 G>C) | Bipolar | 363 | 1 | 0 | 0.0878 | 727 | 1 | 0.0879 |
| | Control | 1060 | 0 | 0 | | 2120 | 0 | |
| | | G/G | G/T | C/T | | G | T | |
| p.Ala444Ser | Schizophrenia | 1040 | 0 | 0 | 1.0000 | 2080 | 0 | 1.0000 |
| (c.1,330 G>T) | Bipolar | 362 | 2 | 0 | **0.0157** | 726 | 2 | **0.0158** |
| | Control | 1060 | 0 | 0 | | 2120 | 0 | |

tested whether SLITRK1 overexpression can affect the distribution of monoaminergic fibers in vivo. Accordingly, expression vectors for inducing the expression of SLITRK1 WT, S330A, A444S, and L422fs with membrane-anchored ECFP (pCAG-SLITRK1-ires-mCFP) were constructed and transfected into the hemilateral somatosensory cerebral cortex through in utero electroporation (Fig. 8h). SLITRK1-L422fs was a frame-shift truncation mutant identified in a family with TS and in patients with TTM[1]. Electroporated mice were immunostained with the anti-NET antibody at P5. We compared the densities of NET-positive fibers with those at the corresponding sites in the contralateral cerebral cortex. Compared with empty vector (pCAG-ires-mCFP) or SLITRK1-L422fs electroporation, SLITRK1-WT and SLITRK1-S330A reduced the NET-positive fiber densities (empty-WT, $P = 0.044$; fs-WT, $P = 0.0035$; fs-S330A, $P = 0.021$); however, SLITRK1-A444S increased the NET-positive fiber densities by 60% (WT-A444S, $P = 0.00028$) (Fig. 8h, i). SLITRK1-L422fs, which was predicted to generate a C-terminal-truncated ECD fragment, did not affect the NET-positive fiber densities (Fig. 8h, i).

The gain-of-function (overexpression) and loss-of-function (KO) findings suggested that SLITRK1 WT could suppress the noradrenergic projections in vivo. The overexpression experiment indicated that A444S and WT had functional differences.

**SLITRK1 ECD physically interacts with L1 family proteins**. To clarify the molecular mechanism underlying the SLITRK1-mediated control of neurite development, we searched for Slitrk1-interacting proteins. We performed an immunoprecipitation assay using mouse brain membrane fractions solubilized in a lysis buffer containing 0.5% Triton-X 100 and anti-SLITRK1 C-

terminus antibody. Next, the co-precipitant was identified using mass spectrometry (Fig. 9a and Supplementary Table 3), and binding was confirmed by pulldown assays using purified recombinant proteins (Supplementary Figs. 8 and 9). As shown in Fig. 9a, we identified Dynamin1 and L1 family molecules, including Neurofascin, L1CAM, and neural cell adhesion molecule (NCAM). Dynamin1 regulates endocytosis or membrane internalization[35,36]. Regarding L1 family proteins, the physical interactions between L1 family-ECDs and SLITRK1 ECD-immunoglobulin Fc domain-fusion protein (SLITRK1-WT-Fc) were quantitatively assessed by quartz crystal microbalance that measures mass variation per unit area by measuring the change in the frequency of a quartz crystal resonator. The addition of aliquots of L1 family-ECDs solution to SLITRK1 ECD-Fc immobilized on a thin plate attached to a crystal resonator showed a concentration-dependent decrease in the frequency, enabling us to determine the kinetic parameters. The $K_D$ values for Neuofascin ECD and L1CAM ECD were 16.6 and 5.47 nM, respectively (Fig. 9b). In this assay system, the $K_D$ value for SLITRK1-WT-Fc and PTPRD, a known binding partner for SLITRK1 at synapses, was 152 nM, which was consistent with previous findings [313 nM for PTPRD and SLITRK1-LRR1-Fc[19]; 58.6 nM for PTPRD and SLITRK3[18]]. The $K_D$ values for Neurofascin-SLITRK1-S330A-Fc and Neurofascin-SLITRK1-A444S-Fc were 59.3 and 14.2 nM, respectively (Fig. 9b). The $K_D$ value for SLITRK1-S330A-Fc was significantly higher than that of SLITRK1-WT-Fc ($P = 0.0045$, $n = 3$ experiments). Furthermore, the binding kinetics, which were reported as $k_{on}$ and $k_{off}$, were significantly higher in SLITRK1-A444S-Fc than in SLITRK1-WT-Fc ($k_{on}$, $P = 3.4 \times 10^{-4}$; $k_{off}$, $P = 2.9 \times 10^{-4}$, $n = 3$ experiments) (Fig. 9b). Contrarily, the L1CAM-binding kinetics was comparable between SLITRK1-WT-Fc and the two variants.

Next, we examined whether Neurofascin co-expression affected the neurite-controlling activity of the SLITRK1. In the in utero electroporation-based neurite assay, Neurofascin co-expression rescued the SLITRK1-induced suppression of the total branch length and number of proximal neurites (30–60 μm from the cell body) (Fig. 9c–e). We also tested the addition of Neurofascin ECD or NCAM ECD to SLITRK1-WT, SLITRK1-S330A, and SLITRK1-A444S electroporated neurons. While Neurofascin ECD increased the neurite numbers of SLITRK1-WT, the minimal effect was seen on S330A or A444S electroporated neurons (Supplementary Fig. 8). Similarly, adding NCAM ECD to WT electroporated neurons increased their neurite numbers with negligible effects on S330A or A444S electroporated neurons (Supplementary Fig. 8). Taken together, the effects of the Neurofascin proteins differed between WT and the two variants, which was consistent with the altered binding affinities of S330A and A444S to Neurofascin ECD (Fig. 9b).

**SLITRK1 suppresses Sema3a-induced endocytosis**. Given the observed binding to Dynamin1 (Fig. 9a and Supplementary Fig. 9), we investigated the role of Slitrk1 on endocytosis. In NGF-treated PC12D cells, SLITRK1 prolonged the endocytosis duration and increased the clathrin-positive vesicle count (Supplementary Fig. 9). Since another Slitrk1-binding molecule, L1CAM, can mediate Sema3a-induced receptor endocytosis[37], we examined whether SLITRK1 affects Sema3a-induced endocytosis. We added Sema3a-Fc to COS7 cell transfectants expressing L1CAM and Neuopilin1 (NRP1, Sema3a receptor) with or without SLITRK1. The endocytosed Sema3a-Fc was detected together with co-endocytosed fluorescent probe FM4-64. In this assay, adding Sema3a-Fc increased FM4-64$^+$NRP1$^+$L1CAM$^+$ endocytosed vesicle counts in the absence of SLITRK1; however, they were suppressed by the presence of SLITRK1 ($P = 0.0033$) (Fig. 9f, g). S330A and A444S reduced the levels of Sema3a$^+$FM4-64$^+$Nrp1$^+$L1CAM$^+$ vesicles comparable to those of WT

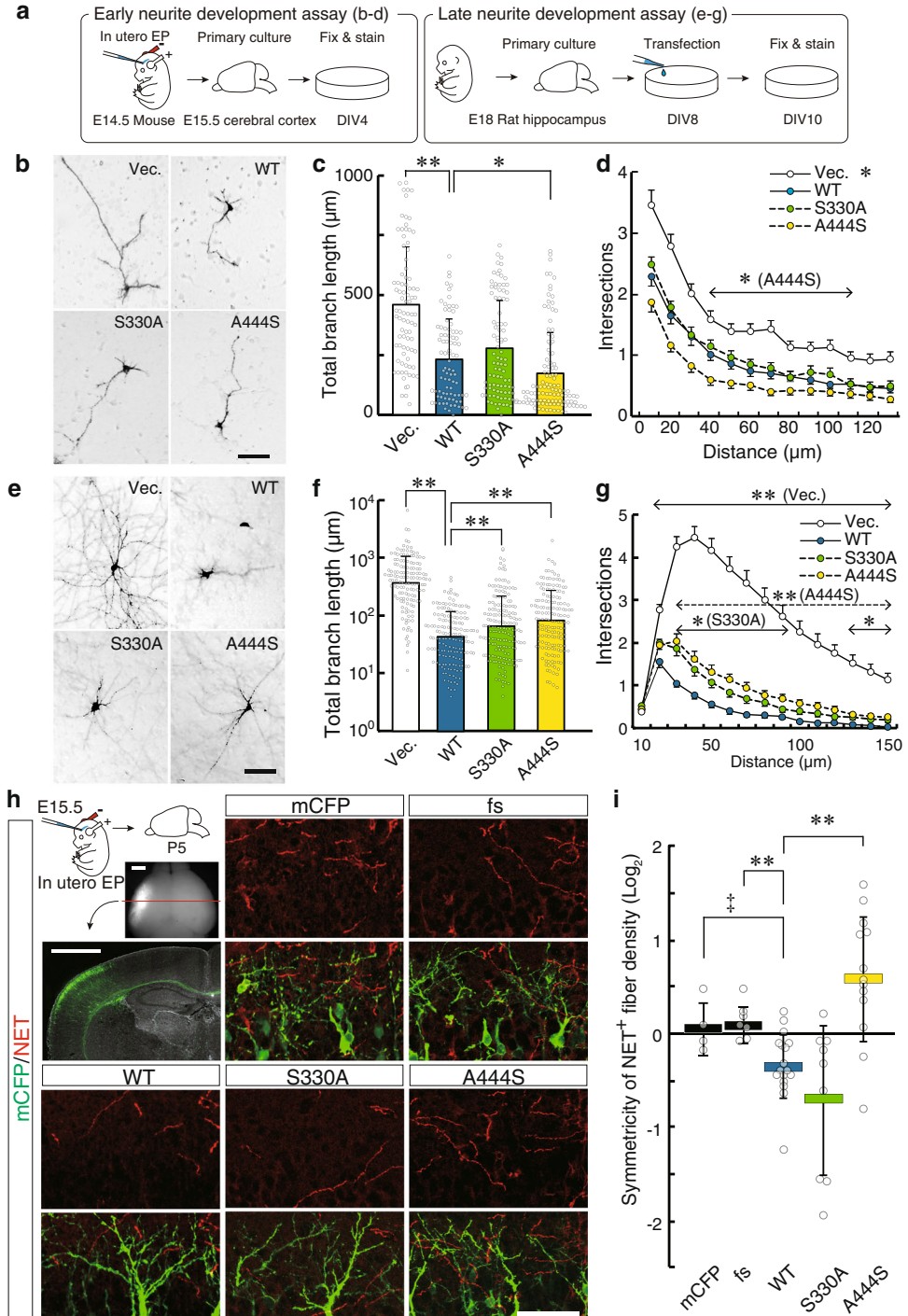

(Fig. 9f, h), which was in agreement with the comparable L1CAM-binding properties between WT and the two variants.

## Discussion
Our findings revealed the following Slitrk1-KO neonatal phenotypes: (1) body weight loss (P3♂), (2) altered vocalization (P4–10♂♀), (3) excessive NA projections in the PFC (P7♂♀), (4) enlarged 5HT varicosity in the PFC (P7♂), (5) increased NA levels in the PFC (P7♂). Regarding subsequent developmental stages, adult Slitrk1-KO-like abnormalities (reduced locomotion and depression-like behavior) were manifested as early as 5W. Further, there was consistent body weight loss only in males after 3W. At 6M, there was enlarged cholinergic varicosity in the PFC.

Determining the sex-dependency of each phenotype could facilitate the elucidation of the causal relationships among the neonatal phenotypes. Excessive NA projections showed the least sex differences, with the other phenotypes showing varying sex differences. The discrepancy between the sex-dependency of excessive NA projections and NA levels in females may be attributed to sex differences in the NA dynamics reviewed in ref. [38]. Specifically, there has been increased attention on sex differences in VMAT2 function[39], where female mice possess greater striatal VMAT2 levels/activity, as well as the sex-dependent role of glucocorticoid receptors in the noradrenergic system[40]. Furthermore, catechol O-methyl transferase (COMT), which is a NA and dopamine metabolizing enzyme, exhibits sexual dimorphism[41].

**Fig. 8 Effects of patient-derived mutations on SLITRK1 function. a–g** In vitro assays. **a** Procedures for the neurite development assay in early or late stages. In the early neurite development assay (**b–d**), cortical neurons were cultured after in utero electroporation to express SLITRK1-WT, SLITRK1-S330A, or SLITRK1-A444S, followed by the total branch length and complexity analyses at DIV4 (Vec., $n = 86$; WT, $n = 90$; S3330A, $n = 97$; A444S, $n = 120$ cells). In the late neurite development assay (**e–g**), hippocampal primary culture and transfected at DIV8, and the analyses were done at DIV10 (Vec., $n = 138$; WT, $n = 157$; S330A, $n = 172$; A444S, $n = 160$ cells). **b, e** Representative images of the neurons. We identified electroporated cells by detecting alkaline phosphatase activity derived from the expression vector. Scale bar, 100 μm. **c, f** Total branch length. Values are presented as mean ± SD. Outliers were removed after Grubbs' test. **d, g** Branch numbers (intersection of Sholl analysis). In (**d**) and (**g**), open circle, empty vector electroporated neuron; cyan circle, SLITRK1-WT; green circle, SLITRK1-S330A; yellow circle, SLITRK1-A444S. Values are presented as means ± SEM (**d, g**); *$P < 0.05$; **$P < 0.01$ in Dunnett's test (**c, f**) or Steel's test (**d, g**) compared with WT transfectants. In (**d**) and (**g**), the statistical tests were performed at each distance, and the distances with significant differences are indicated by double-headed arrows. **h, i** In vivo assay assessing effects of exogenous SLITRK1 on cortical NET$^+$ fiber distribution. In utero electroporation to the hemilateral somatosensory cortex was performed at E15.5 to express SLITRK1-WT, SLITRK1-fs, SLITRK1-S330A, or SLITRK1-A444S with membrane-anchored CFP (mCFP), followed by immunostaining analysis of the NET$^+$ fibers at P5 as illustrated. (GFP, $n = 4$; fs, $n = 7$; WT, $n = 17$; S330A, $n = 9$; A444S, $n = 13$ mice). **h** Representative images. Red, NET; green, electroporated cells were identified by immunostaining for CFP. Scale bar, 50 μm. **i** The layer II/III NET$^+$ fiber density was normalized using that of the contralateral equivalent region. The normalized values were plotted in log$_2$-scale, where zero indicates equal NET + fiber density between the ipsilateral and contralateral regions, and a score below zero indicates the treatment-induced suppression of the NET + fiber density. Values are presented as mean ± SD. *$P < 0.05$; **$P < 0.01$ in Steel's test, compared with WT transfectants. ‡$P < 0.05$ in t-test between GFP and WT transfectants.

Compared with WT mice, male and female COMT$^{-/-}$ mice showed three-fold higher and unchanged cortical dopamine levels, respectively[42], which indicated the existence of a sex-specific compensatory mechanism. Moreover, the involvement of COMT sexual dimorphism in Slitrk1 mice was further indicated by the increase in levels of homovanillic acid (a COMT-dependent dopamine metabolite) in Slitrk1-KO male, but not female, mice (Fig. 4). Accordingly, the excessive NA projections may have caused the other sex-dependent neonatal phenotypes in Slitrk1-KO mice and we have considered these phenotypes from this perspective (Fig. 10a).

The synaptic phenotype could be explained as follows. NA suppresses 5HT secretion through α2 heteroreceptor stimulation[43]. The 5HT varicosity enlargement in the PFC and NAC could be attributed to α2 heteroreceptor on 5HT presynapse (varicosity) suppressing the release and the increase in presynaptic 5HT retention. This idea is further supported by the clonidine injection experiment (Fig. 5). Accordingly, there may have been decreased extracellular 5HT levels in the PFC of Slitrk1-KO mice since tissue 5HT levels remain unchanged despite the increased 5HT varicosity size. Additionally, α2 heteroreceptor suppresses acetylcholine release in the PFC[44]. Therefore, enlargement of ChAT varicosity in the PFC at 6M (Fig. 3) may be well explained by increased NA levels in the PFC of adult Slitrk1-KO mice[24]. Contrarily, the NET$^+$ varicosity size in the PFC of male Slitrk1 KO was larger at P7 but smaller than those of WT at 5W or 6M (Fig. 2). While the increase at P7 can be interpreted as feedback from excessive NA via α2 autoreceptor, the decrease at later stages suggests the presence of some adaptive mechanisms for the excessive NA. As a candidate mediator of such adaptive responses, VMAT2 should be noted because VMAT2 is a critical regulator of presynaptic NA storage in the brain, and VMAT2 expression is dynamically regulated both during development and upon acute and chronic drug exposure[45].

The body weight and vocalization phenotypes are well explained by assuming serotonergic disturbances. Mice lacking 5HT in the brain show severe growth retardation with lower body manifestations at P3[46,47]. Both NA and 5HT are related to altered vocalization in rodents[33,48,49]. Since serotonergic regulation plays a critical role in neonates, the infantile phenotype of the Slitrk1-KO mice may reflect the reduced 5HT levels. Accordingly, it should be noted that the 5HT metabolite (5-HIAA) is increased at the NAC of adult male Slitrk1-KO mice[24], which suggests that the altered 5HT signaling is not limited to the neonatal stage.

Taken together, our findings suggest that the neonatal PFC phenotypes of Slitrk1-KO mice may be associated with excessive

NA projections, which could facilitate further elucidation of the role of NA in refining the PFC status during neonatal development.

We observed significantly reduced Sema3a, Slit2, and Slit3 expression levels in the PFC of Slitrk1-KO mice (Fig. 6a). Among them, Sema3a suppressed NA projections in vitro (Fig. 6b–d). Sema3a signaling involves L1CAM as an NRP1-related signal transducing transmembrane protein[37,50]; moreover, L1CAM increases Sema3a receptor endocytosis[51]. The observed Slitrk1–L1CAM physical interaction and Slitrk1-mediated suppression of Sema3a/NRP1/L1CAM endocytosis suggest that Slitrk1 deprives the Sema3a/NRP1/L1CAM complex of L1CAM and affects the signaling efficacy of Sema3a. Although this hypothesis requires further validation, our results revealed another contact point between Slitrk1 and Sema3a signaling; specifically, Slitrk1-mediated regulation of Sema3a gene expression. Accordingly, Sema3a acts as both the receptor and ligand in the bidirectional regulation of Sema3a signaling[52]. Therefore, the functional linkages among Slitrk1, L1CAM, and Sema3a signaling may be involved in regulating NA projections in developing brains (Fig. 10b, c).

At P4, Slitrk1 was strongly expressed in the superficial layer of the cerebral cortex with dense NA projections (http://developingmouse.brain-map.org/experiment/show/161154659, Supplementary Fig. 4). However, Slitrk1 expression broadly occurs at the same site, including the hindbrain region, which includes LC progenitors. Given the lower sensitivity to Sema3a-mediated suppression in LC neurons derived from Slitrk1-KO mice (Fig. 6c), excessive NA projections could partly result from the loss of cell-autonomous Slitrk1 function in LC neurons. Contrastingly, Slitrk1 ECD exerted neurite-suppressing activity in the same culture (Fig. 6c, d). Together with the fact that Slitrk1 ECD is released from cell membranes by γ/α-secretases[17], the aforementioned finding is suggestive of direct suppression of NA projections in a cell non-autonomous manner (Fig. 10b, c). Consequently, both cell-autonomous and non-autonomous modes could be the molecular basis of the Slitrk1-mediated suppression of NA projections. Additionally, both modes could be mediated or modulated by the physical interaction between Slitrk1 and L1 family since both Neurofascin and L1CAM are strongly expressed in the cortex at P4[53]. The fact that the neurite suppressive effect of SLITRK1 ECD requires Slitrk1 in LC neurons suggests that Slitrk1 and Slitrk1 ECD may compete for the L1 family proteins. However, we cannot exclude other possibilities, such as homophilic interaction via Slitrk1 ECDs or its binding to unidentified targets. There is a need for further studies to elucidate the contribution of each mode to the excessive NA projections in the PFC of Slitrk1-KO mice.

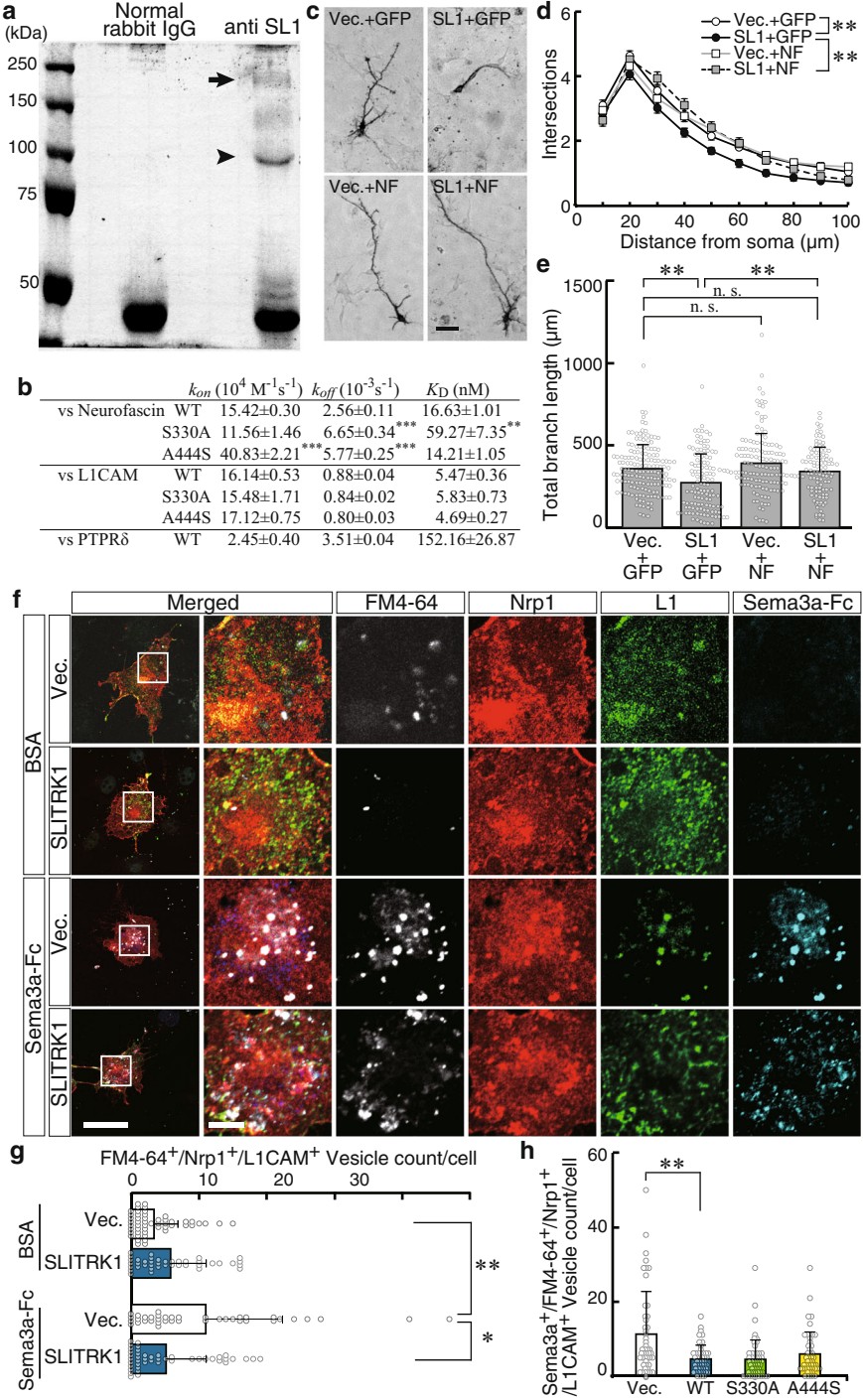

**Fig. 9 SLITRK1 physically interacts with L1CAM family proteins. a** Coomassie brilliant blue-stained sodium dodecyl-sulfate polyacrylamide gel electrophoresis after immunoprecipitation using anti SLITRK1 or normal rabbit IgG. 200 kDa band (*arrow*) and 100 kDa band (*arrowhead*) were excised and subjected for mass-spectrometry analysis (Supplementary Table 3). The blot image was uncropped. **b** $K_D$, $k_{on}$, and $k_{off}$ values of SLITRK1 or its variants-binding to Neurofascin, L1CAM, and PTPRδ are presented as mean ± SD (Neurofascin and PTPRδ, $n = 3$; L1CAM, $n = 6$ experiments). **c** Representative images of cultured cortical neurons overexpressing SLITRK1 and Neurofascin (*SL1 + NF*). *Vec.,* empty vector (negative control for SLITRK1-expressing plasmid); *GFP* (control for Neurofascin). **d** Sholl analysis. *Open circle*, empty vector + GFP ($n = 151$ cells); *filled circle*, SLITRK1 + GFP ($n = 117$ cells); *gray rectangle*, SLITRK1 + Neurofascin ($n = 103$ cells); *open rectangle*, empty vector + Neurofascin ($n = 132$ cells). Values are presented as mean ± SEM. **e** Total branch length. **f–h** Effects of SLITRK1 on L1CAM-mediated endocytosis of the Sema3a receptor Nrp1. Nrp1/L1CAM-expressing or Nrp1/L1CAM/SLITRK1-expressing COS7 cells were incubated in a medium containing FM4-64 (a fluorescein probe for plasma membrane) with Sema3a or BSA. FM4-64⁺/Nrp1⁺/L1CAM⁺ particles (**g**) or FM4-64⁺/Nrp1⁺/L1CAM⁺/Sema3a⁺ particles (**h**) were counted in each cell. **g** Vec. (BSA), $n = 51$; SLITRK1 (BSA), $n = 40$; Vec. (Sema3a), $n = 39$; SLITRK1. (Sema3a), $n = 45$ cells. **h** Vec., $n = 48$; WT, $n = 48$; S330A, $n = 51$; A444S, $n = 48$ cells. Values are presented as mean ± SD (**e, g, h**). *$P < 0.05$; **$P < 0.01$; ***$P < 0.001$ in the Dunnett's test (**b**), Steel's test (**e, g, h**), or Tukey's test (**d**). *Scale bar*, 50 μm (**c, f**, low magnification), 10 μm (**f**, high magnification).

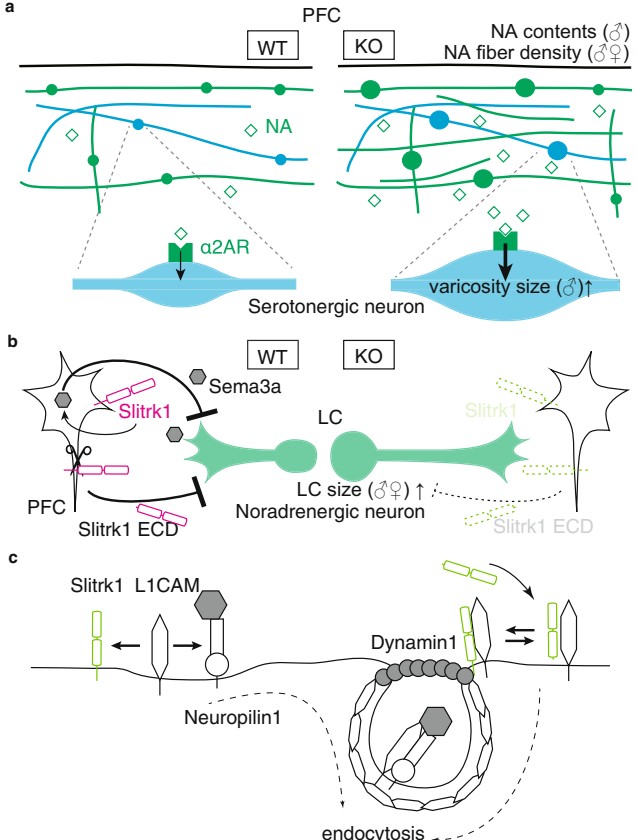

**Fig. 10 Hypothetical mechanisms for Slitrk1-mediated control of monoaminergic fiber development.** The pictures indicate the mechanisms underlying the **a** occurrence of the Slitrk1-KO P7 brain phenotype, **b** Slitrk1-mediated noradrenergic fiber suppression, and **c** Slitrk1-mediated endocytosis regulation.

Our findings suggested the pathophysiological basis of SLITRK1-associated OCRD. Taken together with the increased expression of human SLITRK1 in the developing and mature PFC (Brainspan database, https://www.brainspan.org/rnaseq/search/index.html)[16], altered NA signaling in the PFC could be involved in the altered CSTC functionality in patients with OCRD.

The idea may be supported by accumulating evidence. First, the OCD-like phenotype can be paradoxically yielded by developmental exposure (P9–P16 i.p.) to clomipramine[13]. Second, neonatal clonidine injection (P1–P21) has been shown to cause long-lasting changes in NA sensitivity in rats[54], which suggests that neonatal NA dysregulation affects adult brain functionality. Third, aromatase-KO male mice, who have reduced hypothalamic COMT expression, develop OCD-like behavior; however, aromatase-KO female mice did not present OCD-like behavior or altered COMT expression[55].

Early-onset OCD tends to be more severe, more familial, and male predominant; moreover, it is associated with tic disorders, other comorbidities, and poorer treatment response[12,56]. Taken together, early-onset OCD might partly share altered monoaminergic regulation as a common etiology. Although pharmacological perturbation of 5HT or dopamine could yield OCD animal models[12], the consequences of neonatal NA disturbance remain unclear. There is a need for further studies to clarify the effects of the altered neonatal NA on later neural circuit properties to further elucidate the OCD pathophysiology.

We identified two nonsynonymous mutations (S330A and A444S) that alter the SLITRK1 function. Our findings revealed

another aspect regarding *SLITRK1*, where human genetic studies have been limited to TS and OCD. However, recent genome-wide association studies (GWAS) have demonstrated that *SLITRK1* is related to SCZ[57] and its sensitivity to lurasidone, which is an antipsychotic[58]. Another GWAS on irritable vs. elated mania suggested the presence of genetic loci that affected bipolar disorder subtype discrimination of the 13q31 region, which includes *SLITRK1*[59]. Taken together, *SLITRK1* may be considered an SCZ/BPD-related gene in parallel to its association with OCRD.

Both S330A and A444S functionally alter the SLITRK1 protein as follows. First, both mutants showed altered neurite-controlling abilities in early and/or late neurite development. Second, both mutations impaired SLITRK1 activity that induces inhibitory synapses. Third, A444S and SLITRK1 recruited and repelled the NET-positive fibers, respectively. Fourth, regarding binding to Neurofascin, S330A increased the $K_D$ value while both S330A and A444S altered the binding kinetics. Fifth, Neurofascin ECD abolished the effect of SLITRK1 WT, but not SLITRK1 S330A or A444S, on neurite development. Taken together, both S330A and A444S affect multiple SLITRK1 functions without deleterious effects on stability, trafficking, or ER stress.

Other than the S330A and A444S, function-damaging missense mutations of SLITRK1 have been reported. Ozomaro et al.[6] identified two SLITRK1 missense mutations (N400I and T418S) in 381 patients with OCD and found that N400I impairs the neurite-controlling activity of SLITRK1. Kang et al.[21] reported that both N400I and T418S reduce surface expression in HEK293 cells and increase ER retention in COS7 cells. Accordingly, SLITRK1 N400I and SLITRK1 T418S impair dendritic targeting and reduce synaptogenic activity[21]. The mutation effects of S330A and A444S may be more restricted than those of OCD-derived mutations (N400I and T418S). Specifically, A444S showed opposing activity to NET-fiber repelling activity of SLITRK1 WT (Fig. 8h, i), which suggests that A444S possesses the gain-of-function property. These results raise an interesting question regarding whether SLITRK1 protein variants have disease-specific functional alterations, which could be confirmed by further examination of missense mutations and a more comprehensive comparative analysis.

SLITRK1 S330A mutation is intriguing from the perspective of human evolution. A recent genome sequence analysis of two archaic human groups (Neandertals and Denisovans) revealed SLITRK1 A330S mutations in *Homo sapiens* ancestor after it diverged from the common ancestor of Neandertals and Denisovans (804,000 years ago)[60–62]. *SLITRK1* has been included in eight genes strongly conserved in the ancestral state in primates but derived from modern humans; moreover, its function is related to brain function or nervous system development[61]. Another study included SLITRK1 in three behavioral category genes, with amino acid-changing substitutions being overrepresented in the modern human lineage after splitting from the archaic human groups[60]. Taken together with the observed molecular functional differences of SLITRK1 S330A, we propose that SLITRK1 could be associated with acquiring brain functions unique to modern humans.

## Methods

**Animals.** All animal experiments were approved by Animal Experiment Committees at the RIKEN Brain Science Institute and Animal Care and Use Committee of Nagasaki University. Further, they were conducted following the guidelines for animal experimentation at RIKEN and Nagasaki University. Slitrk1-deficient mice were established and genotyped as described[24]; moreover, they were backcrossed to C57BL/6J mice for more than ten generations before starting the experiments. Both male and female newborns to 6 months old of age were used for experiments. We purchased CD-1 mice and Sprague−Dawley rats from Japan SLC and CLEA Japan. Data for the body weight curve (Fig. 1a) were collected before other experiments using the animals. Mostly common animals were used between immunostaining

analyses for two distinct regions (Figs. 2 and 3). The other experiments were carried out using experimentally naïve mice.

**Study design**. The sample sizes for each experiment were determined such that the power and significance in the two-sided test were 80% and 5%, respectively, according to Festing[63]. However, the number of samples from the animals was minimized empirically. All data were collected in sample-order-randomized manners. A behavioral test was carried out in an animal identity-blinded manner. Neurite analyses were performed by experimenters who were blinded to the specimen identity. The experiments for assessing the SLITRK1 mutations were carried out in a plasmid identity-blinded manner.

**Cell lines**. COS7 cells, HEK293T cells, and PC12 cells were obtained from RIKEN Bioresource Center Cell Bank (https://cell.brc.riken.jp/). PC12D cells were provided by Dr. Shinichi Hisanaga (Tokyo Metropolitan University) and were originally developed in Katoh-Semba et al. [64]. No commonly misidentified cell lines were used in this study. The cell lines were not tested for mycoplasma contamination. None of the cell lines used were authenticated.

**Behavioral analysis**. USV was recorded as described[65]. In this assay, postnatal day 4–14 (P4–P14) mice were used. The mouse pups were isolated from their dams for 5 min. We analyzed recorded sound files using SASLab Pro Recorder Software (Avisoft Bioacoustics, Berlin, Germany). The number of calls, as well as the duration, interval, start time, end time, peak frequency (start), peak amplitude (start), peak frequency (end), peak amplitude (end), peak frequency (center), and peak amplitude (center) of each call, were analyzed. The USV was categorized as Flat in case the difference between the start and end frequencies was <500 Hz. The other recordings were categorized as Upward or Downward. Additionally, in case the center frequency was higher than the average start and end frequency +250 Hz, we categorized the USV as Convex. Similarly, in case the center frequency was lower than the average −250 Hz, we categorized the USV as Concave.

Open field test and forced swimming test was performed as described[24]. For open field test, each mouse was placed in the center of an open-field apparatus (50 × 50 × 40 (H) cm) and then allowed to move freely for 10 min. Distance traveled was measured and analyzed using Image J OF4 (O'Hara, Tokyo, Japan). Total distance traveled, percentage of time in the open arms and percentage of open arm entries were measured as indices. Data were collected and analyzed using Image J EPM (O'Hara). For the forced swimming test, each mouse was placed for 7 min in a glass cylinder (30 cm high, 10 cm in diameter) containing 10 cm of water maintained at 22–25 °C. The duration of immobility was recorded.

Behavioral analysis of developing mouse pups as described by Fox[28] and Heyser et al. [29]. The list of *test subject*: experimental procedure (range of value or scoring criteria) are as follows: *righting reflex*: place the pup gently onto its back and record the time for the subject to turn over onto its belly with upper limit of 60 s (0–60 s); *negative geotaxis*: place the pup on a 45° inclined plane with its head pointing down the incline, and record the latency to turn around and crawl up the slope, up to a maximum of 60 s (0–60 s); *placing response*: suspend the pup in the air by grasping the animal gently around the trunk, bring a thin metal bar into contact with the back its paws, and record whether the pup raises and places one of paws on the surface of the bar (0, no reaction; 1, weak reaction; 2, strong reaction); *grasp response*: stroke the paw with a wooden toothpick, and record whether the reflex is present (0, no reaction; 1, weak reaction; 2, strong reaction); *auditory startle response*: expose the pup to acoustic stimuli by operating a clicker (20 kHz, 90 dB, O'hara) at a distance of 25–30 cm above each pup, and record whether or not the animal suddenly extends the head, fore and hind limbs (0, no reaction; 1, weak reaction; 2, strong reaction); *tactile startle response*: gently apply a puff of air to the pup by pushing bulb of a polyethylene 10 mL pipette, and record whether or not the animal suddenly extends the head, fore and hind limbs (0, no reaction; 1, weak reaction; 2, strong reaction); *bar holding*: lift the pup by the trunk and bring it up close to a thin metal bar of 6 mm diameter, allow it to grab hold with its front paws, and record how long the pup holds on the bar after being released, up to a maximum of 10 s (0–10 s); *vertical screen test*: place the pup at one end of a wire mesh screen, rotate the mesh to a vertical position to assess whether the animal stays holding on the mesh (0, falling off before rotation; 1, falling off after rotation; 2, staying); *level screen test*: place the pup onto a piece of metal mesh, gently drag the pup in a horizontal direction by the tail, and record whether the pup can hold onto the screen (0, no holding; 1, weak holding; 2, strong holding); *quadruped locomotion-pivoting*: place the pups on flat surface, record whether the animal show a pivoting motion, in which the pup makes broad swipes with the paws, producing a paddling motion that result in turning (0, absent; 1, present); *quadruped locomotion-crawling*: place the pups on flat surface, record whether the animal show a crawling motion, in which the paddling movements of the paws result in animal dragging itself forward or pushing itself backward (0, absent; 1, present).

**Immunofluorescence staining and image analysis**. Mice were anesthetized by inhalation of isoflurane, and cardiac perfusion was performed with 4% paraformaldehyde and 0.1 M sodium phosphate (pH 7.4). Excised brains were fixed in the same fixative for 1 h at 20–25 °C with gentle agitation. The brains were trimmed and cryoprotected with 20% sucrose in phosphate-buffered saline [PBS(−)] at

4 °C overnight. After adding the OCT compound (Sakura Finetek) to 60% (v/v), the tissue blocks were placed in the resultant embedding medium for 15 min with gentle agitation. The blocks were then placed in Cryomold (Sakura Finetek) with the embedding medium and rapidly frozen on dry ice or pre-chilled aluminum blocks at −80 °C. Cryosection was performed using a CM3050 cryostat (Leica Biosystems) at a thickness of 25 μm. After sectioning, the sections were blocked with 2% normal goat serum and 0.1% Triton X-100 in PBS and reacted with primary antibodies at 4 °C for 1–4 days, followed by appropriate fluorescence-labeled secondary antibodies at room temperature for 1 h. FOR SERT immunostaining, the sections were blocked with 20% Block ACE (Dainippon Pharmaceutical), 5% normal goat serum, and 0.2% Triton X-100 in PBS(−), and the first antibody was diluted with 5% Block ACE and 0.2% Triton X-100 in PBS(−). We used the following antibodies at the indicated dilution: anti-NET (NET05-01, Mab technologies, Inc., 1:5000), anti-ChAT (AB144P, Millipore, 1:300), anti-SERT (HTT-Rb-Af560, Frontier Institute, 1:300), anti-DAT (MAB369, Millipore, 1:1000), anti-green fluorescent protein (GFP) (#598, MBL, 1:1000), anti-VGLUT1 (NeuroMab, N28/9, 1:500), anti-VGAT (#131 003, Synaptic Systems, 1:1000), anti-microtubule-associated protein 2 (MAP2) (AB5622, Millipore, 1:1000), anti-HA (3F10, Sigma, 1:1000), and anti-phospho-extracellular signal-regulated kinases (SC-7383, Santa Cruz, 1:200). Staining was performed using Alexa fluorescence dye-conjugated secondary antibodies (Thermo Fisher Scientific), with the images being obtained using a confocal microscope (Olympus FV-1000 or Zeiss LSM-800). The beaded structure and dendrite length were measured using ImageJ software. To determine the beaded structure, the Subtract Background command was run with the rolling ball radius 20.0 pixels option, with the beaded structure being defined as a size between 4 and 20 pixels and circularity between 0.3 and 1.0. Further, the Skeletonize command was run to determine the dendrite length.

**Monoamine content analysis**. At P7, brains were collected immediately after decapitation and 150 μm-thick frozen coronal sections were prepared. The medial PFC, striatum, and dorsal raphe were dissected using a biopsy needle (1 mm diameter; Biopsy Punch, Kai Medical). Six punches were collected from each region and stored at −80 °C until extraction. For each specimen, 100 μL extraction buffer (0.1 M perchloric acid, 200 mM isoproterenol, 0.1 mM EDTA, pH 8.0) was added and homogenized using a sonicator (VCX 130, Sonics & Materials). Subsequently, 15 μL of the homogenate was removed for protein quantification assay. The remaining homogenate was incubated on ice for 30 min, followed by centrifugation at $20,000 \times g$ for 15 min at 0 °C. Supernatants were collected, followed by the addition of 8.8 μL of 1 M sodium acetate per 100 μL supernatant and filtration through a 0.22 μm filter (Ultrafree, Millipore). Specimens were frozen at −80 °C until high-performance liquid chromatography analysis.

We analyzed NA, MHPG, dopamine, 3,4-dihydroxyphenylacetic acid, homovanillic acid, serotonin (5HT), and 5-HIAA using HPLC with an electrochemical detector (HTEC-500, Eicom, Kyoto, Japan). These substances were separated on a SC-5ODS column (Eicom). Isoproterenol was used as an internal standard. The flow rate was maintained at 0.5 mL/min. We used a pure graphite working electrode set at a potential of 750 mV with an Ag/AgCl reference electrode. The mobile phase contained 0.1 M sodium acetate/0.1 M citrate buffer (pH 3.5), 17% methanol, 1-octanesulfonic acid 190 mg/L, and EDTA·2Na 5 mg/L. Chromatographs were acquired using the software PowerChrom (EPC-500, Eicom, Kyoto, Japan).

**Clonidine/clomipramine administration**. Clomipramine or clonidine was dissolved in saline. Clomipramine (15 mg/kg/day), clonidine (0.1 mg/kg/day), or saline was administered to C57BL/6J pups intraperitoneally once a day during postnatal days 3–6. The mice were fixed and immunostained as above.

**Quantitative PCR**. Brains were removed from P3 or P7 mice, frozen on dry ice, and sectioned into 150 μm thicknesses. Six biopsy punches were obtained from six sections that included the PFC area (see Supplementary Fig. 7). Trizol was used for total RNA preparation, and to remove genomic DNA contamination, the specimens were digested with DNaseI. cDNA was synthesized from 150 ng of total RNA and gene expression was quantified with PowerUP SYBR green Master Mix (Thermo Fisher). Primer pairs listed in Supplementary Table 4 were used for quantitative PCR. QuantStudio 12K flex real-time PCR system (Thermo Fisher) was used for this experiment. Each gene expression was normalized with GAPDH expression. Specimens were duplicated in each PCR, three independent PCR were performed for quantification, and an average of six results was calculated for one mouse gene expression.

**DNA constructs**. A SLITRK1 expression vector was constructed using a pEF-ires-alkaline-phosphatase plasmid[15] and a pCAG-SLITRK1-ires-ECFP expression plasmid with ECFP modified to have a GAP43-derived membrane anchor signal sequence. Single nucleotide substitution mutations were generated following a PCR-based method using the Quick Change II Site-Directed Mutagenesis Kit (Agilent Technologies). To produce Fc-fusion protein, we cloned SLITRK1 WT, S330A, and A444S ECDs (amino acid number 1-616 in NP_443142) into immunoglobulin Fc domain fusion protein expression vector pEF-Fc, which was donated by Dr. Yoshihiro Yoshihara, RIKEN BSI. We purchased the Neurofascin expression

vector from addgene (http://www.addgene.org, plasmid # 31061 Neurofascin-186-HA). ER stress analysis was performed using pERAI-Luc vector system (Trans-Genic, Fukuoka).

**Neurite and synapse formation assay in primary neuron cultures**. We performed an in-utero electroporation-based assay as described by Abelson et al. [1]. We electroporated pregnant CD-1 mice at embryonic day (E) 15 and isolated cortical neurons at E16. Subsequently, cultured neurons were fixed and alkaline phosphatase-stained at 4 DIV.

For the in vitro transfection-based assay, we obtained hippocampal neurons from the brains of E18 SD rats. Neurons were cultured in Neurobasal Medium supplemented with B27-supplement, GlutaMAX, and an antibiotic–antimycotics cocktail (Thermo Fisher Scientific). To examine dendrite elongation activity, we transfected a SLITRK1-alkaline phosphatase expression plasmid or variant into cultured neurons ($4.4 \times 10^4$ cells/100 mm$^2$) at DIV 8 using Lipofectamine 2000. The cells were fixed and stained using alkaline phosphatase at DIV 10. We purchased recombinant human Neurofascin ECD and NCAM ECD proteins from R&D Systems.

We cultured LC neurons as described by Masuko et al. [66], with some modifications. Brains were collected from C57BL/6J mice at P0 and embedded in 2.5% agar dissolved in Hanks balanced salt solution (HBSS). Next, they were sectioned using Vibratome (LinearSlicer Pro7, Dosaka-EM) at 300-µm thickness. The LC-containing area was dissected using fine forceps, collected into ice-cold HBSS, and dissociated using Trypsin-EDTA (0.05%, Thermo Fisher) for 10 min at 37 °C. Tissue debris was removed using Cell Strainer (70 µm, Corning), with subsequent spreading onto the PDL-coated cover glass at a density of $1-3 \times 10^3$ cells/cm$^2$ and culturing using Neurobasal supplemented with B27, GlutaMax, and antibiotic/antimycotic cocktail (Thermo Fisher). We added recombinant mouse Semaphorin3a-FC (5926-S3, R&D) or SLITRK1 ECD (3009-SK-050, R&D) protein (100 ng/mL each) at DIV2, followed by fixing at DIV5. Cells were fixed using 4% PFA and 4% sucrose, followed by immunostaining using anti-tyrosine hydroxylase (TH, Millipore #AB152, 1:1000) and NET (Mabtechnologies, #NET05-1, 1:1000) antibody. Images were obtained through confocal microscopy (Zeiss LSM-800).

Experimenters who were blinded to the specimen identity performed neurite analyses using Neurolucida (MBF Bioscience). The total branch length and branch complexity (Sholl analysis) were measured from the traced images. Experiments were conducted in triplicates at a minimum. The mean values of three independent wells were obtained in a single experiment. Experimental numbers were considered as $n$ in statistical analyses.

**Participants and mutation screening of SLITRK1**. We performed a resequencing analysis of *SLITRK1* in 1040 (male, 515; female, 525) patients with SCZ, 364 (male, 180; female, 184) patients with BPD, and healthy controls (male, 454; female, 606). All the subjects were recruited from the Honshu area of Japan (the main island of Japan), where the population falls into a single genetic cluster[67]. Using a subset of subjects we previously showed that population stratification is negligible in our samples[68,69]. Best-estimate lifetime diagnosis of patients was made by direct interview with at least two experienced psychiatrists, according to DSM-IV criteria, and using all available information from medical records, hospital staff, and family informants. Patients were recruited by multiple research institutes/hospitals in Japan. Control subjects were recruited from hospital staff and company employees documented to be free from psychoses and were further interviewed by experienced psychiatrists to exclude any psychiatric disorders. All healthy controls were psychiatrically screened using unstructured interviews. All healthy controls and patients provided written informed consent for study participation after receiving an explanation regarding the study protocols and objectives. The study was approved by the ethics committees of RIKEN and all participating institutes and was conducted in accordance with the Declaration of Helsinki. All protein-coding exons and flanking introns were re-sequenced.

**Preparation of the SLITRK1 ECD**. We prepared SLITRK1 ECD-Fc fusion protein as follows. First, HEK293T or COS7 cells were transfected using a SLITRK1 expression plasmid and maintained in a serum-free medium (Opti-MEM) for 3–4 days. The culture medium was centrifuged ($1000 \times g$, 10 min) and filtered through a 0.22-µm syringe filter unit (Millipore) to eliminate cell debris. The secreted SLITRK1 ECD in the culture medium was collected using Protein-A-agarose (20334, Thermo Fisher). The eluted protein was dialyzed against PBS(-) using Slide-A-lyzer (66453, Thermo Fisher), followed by concentration using an ultrafiltration unit (Amicon Ultra, MWCO 50,000, Millipore). The purified protein was stored at −80 °C until binding assay.

**Immunoprecipitation and identification of SLITRK1-binding proteins**. The mice were euthanized and whole brains were surgically removed. The brains were homogenized using a glass-Teflon homogenizer for seven strokes at 700 rpm in 9 mL of the homogenizing buffer (0.32 M sucrose, 0.5 mM HEPES–KOH pH 7.2) per g of the original wet weight of the brain. The homogenate was centrifuged at $1000 \times g$ for 10 min. Next, the supernatant was centrifuged at $10,000 \times g$ for 30 min, with the obtained pellet being lysed using PBS-containing 0.5% Triton X-100 for 1 h. After centrifugation at $10,000 \times g$ for 30 min, the supernatant was collected,

followed by the addition of anti-SLITRK1 rabbit antibody[24] or control normal rabbit IgG. After overnight (4 °C) incubation, we collected IgG and co-precipitant using protein G-conjugated agarose beads. The precipitates underwent sodium dodecyl-sulfate polyacrylamide gel electrophoresis and were detected through Coomassie brilliant blue (CBB) staining or immunoblotting analysis. The CBB-stained band was excised, digested using an in-gel digestion protocol, and analyzed through mass spectrometry using MALDI-TOF MS (Applied Biosystems). Mass spectrometry analysis was done by the Support Unit of Bio-Material Analysis, RIKEN Center for Brain Science Research Resources Division.

**Gravimetric measurements with a biosensor quartz crystal microbalance**. Real-time protein–protein interactions were detected using the AffinixQ system (Initium Inc., Tokyo, Japan), which is a quartz crystal microbalance sensor device. Briefly, we used AT-cut quartz crystals coated with a thin gold surface layer at a fundamental frequency of 27 MHz. Immediately before use, we cleaned the gold surface of the quartz resonator using piranha solution ($H_2SO_4$:30% $H_2O_2$ = 3:1) for 5 min, followed by thorough washing with double-distilled water. Anti-human IgG Fc-HRP (Jackson ImmunoResearch) was applied to the resonator for 30 min to yield a layer for direct immobilization. Subsequently, the resonator was rinsed using interaction buffer (50 mM Tris–HCl pH 7.6, 15 mM NaCl, 140 mM KCl, 0.5 mM MgCl$_2$, 2 mM CaCl$_2$) and immersed in 2 mL interaction buffer. Protein–protein interactions were determined from the frequency changes ($\Delta F$ in Hz) resulting from mass changes on the electrode at the sub-nanogram level upon application of a small volume of protein solution. Based on the Sauerbrey formula, a 1-Hz decrease is calculated as an interaction of 30.38 pg of a molecule with the biosensor. For all immobilizations of the SLITRK1 ECD-Fc recombinant proteins (WT, S330A, and A444S) on the resonator, approximately 200 femtomoles of SLITRK1 ECD-Fc were immobilized as an absolute amount (a 600-Hz decrease). All experiments were conducted at $25 \pm 1$ °C. All sensorgram data revealed the $\Delta F$ value following the association phase. For kinetic analysis, frequency changes induced by each concentration of human Neurofascin ECD-His$_6$ (8208-NF-050, R&D Systems), L1CAM ECD-His$_6$ (777-NC-100, R&D Systems), or human PTPRD ECD-His$_6$ (ACROBiosystems) were curve-fitted to the formula $\Delta F = \Delta F_{t=\infty}\,(1 - e^{(-1/\tau)t})$; further, the $1/\tau$ value was plotted for each concentration of added protein. Additionally, $1/\tau$ and $K_D$ represent $k_{on}[X] + k_{off}$ and $k_{off}/k_{on}$, respectively, where $X$ is the concentration of the added protein.

**In vitro Semaphorin3A-induced endocytosis assay**. COS7 cells were seeded at a density of $5 \times 10^3$ cells/cm$^2$ on collagen IV-coated cover glass 24 h before transfection. Total 1.3 µg of DNA was mixed for each 1 mL culture medium. Nrp1 (pCherry-mNrp1, #21934, addgene), L1CAM (phL1A-pcDNA3, #12307, addgene), and SLITRK1 or its mutant (pEF-SLITRK1-ires-alkaline phosphatase or its derivatives) plasmids were mixed at a ratio of 2:3:3. Transfection was done using polyethyleneimine (Polysciences, #23966)[70]. Five hours after transfection, the medium was changed to fresh DMEM (10% FBS, 1% antibiotic–antimycotic [15240062, Gibco]). Twenty-four hours after transfection, 10 µg/mL of recombinant Sema3a-Fc (5926-S3, R&D) and 5 µg/mL of FM4-64 dye (F34653, Thermo Fisher) were added to the culture medium and incubated for 30 min at 37 °C in 5% CO$_2$ incubator. Cells were washed with a conditioned medium and with cold PBS(−) sequentially and were fixed with ice cold PFA (4% PFA, 100 mM NaCl, 100 mM sodium phosphate, pH 7.0) for 30 min on ice. Cells were washed with PBS(−) and incubated in blocking buffer (2% normal goat serum, 3% BSA, 0.1% Triton X-100 in PBS[-]). L1CAM was detected with anti L1CAM (MAB5272, Chemicon, 1:1000) and Alexa488-conjugated anti-rat IgG. Sema3a-Fc was detected with Alexa633-conjugated anti-mouse IgG. Cell images were obtained using a confocal microscope (Zeiss LSM-800) and analyzed by ImageJ.

**Expression and stability analysis**. SLITRK1 WT and the mutants' expression vector (pEF-SLITRK1-ires-alkaline phosphatase or its derivatives) were transfected using Lipofectamine 2000 (Thermo Fisher) to PC12 cells that were seeded on Collagen IV (Nitta-gelatin) coated plate ($2.5 \times 10^4$ cells/cm$^2$) 24 h before transfection. Twenty-four hours after transfection, the cells were washed using PBS(−) and lysed SDS buffer (2% SDS, 62.5 mM Tris–HCl, pH 6.8, 0.001% bromophenol blue, 10% glycerol, 1% 2-mercaptoethanol). The samples were electrophoresed on 10% polyacrylamide gel. The proteins were transferred to an Immobilon-P membrane (Millipore) and probed with an anti-SLITRK1 rabbit antibody (1:2000) and anti-actin antibody (Sigma-Aldrich, A2066, 1:5000). Horseradish peroxidase conjugated secondary antibodies were used for detection by a chemiluminescence substrate (ECL Plus, GE Healthcare). Densitometric measurement of SLITRK1 and actin protein was done using ImageJ software.

**Luciferase reportor assay for ER stress**. ER stress analysis was performed using pERAI-Luc vector system (TransGenic, Fukuoka). SLITRK1 expression vector, pEF-Renilla luciferase vector, and pERAI-Luc were co-transfected to COS7 cell using Lipofectamine 2000. Cells were harvested after 24 h incubation, and luciferase activity was measured by a Dual-Luciferase reporter assay system (Promega) and a luminometer (Berthold LB9506). The luciferase activity values were normalized with those of Renilla luciferase activity. The values of three independent

wells were obtained in a single experiment, and the average value for WT was defined as 1 in each experiment.

**Subcellular distribution analysis**. For the cell surface labeling assay (Supplementary Fig. 6d, e), N-terminus HA-tagged SLITRK1 WT or mutants (pCAG-HA SLITRK1 or its derivatives) were transfected to COS7 cells seeded on Collagen IV-coated coverglass by Lipofectamine 2000. After 2 days of incubation, an anti-HA rat antibody (3F10, Sigma-Aldrich, 1:100) was added to the medium and incubated for 10 min. Cells were washed once with PBS(−) and fixed with cold 4% paraformaldehyde and 0.1 M sodium phosphate (pH 7.4) for 30 min on ice. Cells were washed with PBS(−) three times and blocked with 3% bovine serum albumin (BSA), 0.1% Triton-X100 in PBS(−) for 30 min on ice. Anti-HA rabbit antibody (Sigma-Aldrich, H6908, 1:1000) was reacted at 4 °C overnight. Cells were washed with PBS(−) and the bound antibodies were detected by reacting with anti-rat Alexa 488-conjugated antibody (1:1000) and anti-rabbit Alexa 594 conjugated (1:1000) at 20−25 °C for 90 min. After staining, cells were observed under a confocal microscope (LSM-800, Zeiss) and analyzed by ImageJ.

For the MAP2 overlapping assay (Supplementary Fig. 6g, h), rat hippocampal neurons from CD-1 mice at E16.5 were cultured. HA tagged SLITRK1 or mutants were transfected by lipofectamine 2000 at DIV 7. On DIV 11, neurons were washed once with PBS(−) and fixed with cold 4% paraformaldehyde, 0.1 M sodium phosphate (pH 7.4), 100 mM NaCl for 30 min on ice. Cells were incubated in a blocking buffer (3% BSA, 2% normal goat serum, 0.1% TritonX-100 in PBS(−)) and reacted with first antibodies (anti-HA antibody, 3F10, 1:1000; anti MAP2 antibody, AB5622, Millipore, 1:1000) for three nights. Isolated neuron images were taken by confocal microscope (LSM-800) and HA and MAP2 positive areas were quantified by ImageJ.

**In vitro secretion assay**. pEF-SLITRK1-WT or mutants expression vector was transfected to COS7 by lipofectamine 2000, and 2 h after, the medium was changed to Opti-MEM. After 2 days of incubation, the conditioned medium was collected and centrifuged at $1000 \times g$ for 10 min. The supernatant was condensed to 1/50–1/100 volume by ultrafiltration (AmiconUltra M.W.C.O 50 kDa, Millipore) and added 1/5 volume of ×6 SDS buffer. Fractions equivalent to one-tenth of the samples were analyzed by immunoblotting and secreted SLITRK1 was probed by anti SLITRK1 ECD antibody (R&D, AF3009, 1:5000).

**Synapse formation assay**. For the heterotopic synapse induction assay (Supplementary Fig. 7), HEK293T cells ($1.3 \times 10^4$ cells/100 mm$^2$) transfected with SLITRK1 expression plasmids (pCAG-SLITRK1-ires-EGFP or its derivatives) were overlaid onto the cultured hippocampal neurons ($2.2 \times 10^4$ cells/100 mm$^2$) at DIV 15. Additionally, cells were fixed using 4% paraformaldehyde containing 4% sucrose for 30 min on ice at DIV 17. The cells were incubated with anti-VGAT mouse antibody (1:1000, Millipore) or anti-VGLUT1 rabbit antibody (1:500, Synaptic Systems) in a blocking buffer (5% normal goat serum, 3% bovine serum albumin, 0.1% Triton X-100 in phosphate-buffered saline [PBS][−]). Immuno-positive puncta were analyzed through ImageJ particle analysis.

**Pulldown assays**. For the assay using SLITRK1 ECD-Fc and brain lysate (Supplementary Fig. 8b), the brain lysate was prepared as described in Immunoprecipitation and identification of SLITRK1-binding proteins. SLITRK1 ECD-Fc (10 μg) or IgFc (2.5 μg) protein was added to tubes containing equal amount of the brain lysate. The mixture was incubated at 4 °C for overnight with gentle agitation. After the incubation, Protein A-conjugated agarose beads (Thermo Fisher, #20334) were added, and after 2 h incubation at 4 °C with gentle agitation, beads were washed with 1 mL of 0.5% Triton X-100 in PBS(−) for three times. Precipitants were eluted with SDS buffer and analyzed with immunoblotting. The following antibodies were used for detection: Anti L1CAM (MAB5272, Chemicon, 1:1000), anti Neurofascin (A12/18, NeuroMab, 1:10,000), and NCAM (MAB310, Chemicon, 1:1000).

For the assay between SLITRK1 ECD and L1CAM-Fc (Supplementary Fig. 8c), 40 pmol of L1CAM-Fc (R&D, #777-NC) as a bait protein and 8 pmol of SLITRK1 ECD (R&D, #3009-SK) were mixed in 500 μL of interaction buffer (10 mM HEPES pH 7.4, 2 mM CaCl$_2$, 1 mM MgCl$_2$, 0.5 % Triton X-100) for 30 min on ice. After the incubation, Protein A-conjugated agarose beads (Thermo Fisher, #20334) were added, and after 2 h incubation at 4 °C with gentle agitation, beads were washed three times with interaction buffer, and eluted with SDS buffer. Precipitants were analyzed with immunoblotting using anti SLITRK1 ECD antibody (R&D, AF3009, 1:5000) or anti human IgG antibody (SouthernBiotech, #2040-01, 1:5000, for the detection of L1CAM-Fc).

For the assay between SLITRK1 ECD-Fc and Neurofascin (Supplementary Fig. 8d), 10 pmol of SLITRK1 ECD-Fc or IgFc protein were captured on Protein A-conjugated agarose beads in 500 μL of interaction buffer (10 mM HEPES, pH 7.4, 2 mM CaCl$_2$, 1 mM MgCl$_2$, 0.1% Triton X-100) for 30 min at 4 °C. Next, 7.5 pmol of Neurofascin 155-His protein (R&D, #8208-NF-050) was added and gently agitated for 1 h at 4 °C. Beads were washed three times with the interaction buffer, and eluted with SDS buffer. Precipitants were analyzed with immunoblotting, and anti-His tag antibody (Santa Cruz, H-15, 1:5000) was used to detect Neurofascin. Anti-human IgG antibody was used to detect SLITRK1 ECD-Fc.

**Immunoprecipitation for Slitrk1–Dynamin1 interaction**. Immunoprecipitation was performed as described in the "Methods" section. To detect Dyamin-1, anti Dynamin1 antibody (MAB5402, Chemicon) was used at a dilution of 1:5000.

**Time-lapse imaging of endocytosis in PC12D cells**. NGF-induced endocytosis was observed in PC12D cells[64]. PC12D cells were maintained in Dulbecco's modified Eagle's medium supplemented with 20% horse serum and 5% fetal bovine serum. The cells were seeded on collagen-coated cover glass at 24 h before observation. mCherry-tagged clathrin and SLITRK1 expression vectors (pEF-SLITRK1-WT) were co-transfected at 24 h after plating using Lipofectamine 2000. The cover glass was mounted on a custom-made perfusion chamber. mCherry-clathrin-derived fluorescent signals were observed using a confocal microscope (FM-1000, Olympus) equipped with a heated stage for a few minutes after the addition of NGF (100 ng/mL) to the chamber. Images were obtained at 1.1-s intervals and analyzed using ImageJ software after noise reduction using the Gaussian blur and subtract background commands. Clathrin vesicles were traced using an ImageJ plugin, Mosaic (https://sbalzarini-lab.org/docs/MosaicSuiteDoc/index.html).

**Statistics and reproducibility**. Statistical analysis was carried out by using Microsoft Excel (Microsoft), BellCurve for Excel (Social Survey Research Information), and SPSS statistical package (version 16, SPSS Inc.). Data were presented as mean ± standard deviation (SD) unless otherwise stated. Between-group differences were analyzed using Student's unpaired two-tailed $t$-test, Welch's unpaired two-tailed $t$-test, or the Mann–Whitney $U$-test. Differences in one-to-many comparison were analyzed by ANOVA with Dunnett's or Steel's test. Two-way or repeated-measures ANOVA was performed to examine the influences of two independent categorical variables. Differences in allele frequency were analyzed by $\chi^2$ test. Multiple test correction was done by Benjamini–Hochberg (BH) post-hoc tests. Outliers were removed after Grubbs' test for monoamine content analysis (Fig. 4) and total branch length analysis in early neurite development assay (Fig. 8c). All attempts at replication were successful.

**Reporting summary**. Further information on research design is available in the Nature Research Reporting Summary linked to this article.

## Data availability

All data generated during this study are included in this published article, its Supplementary Information, and its Supplementary Data 1 files. SNPs have been deposited in NCBI dbSNP (https://www.ncbi.nlm.nih.gov/snp/), and accessions are listed in Supplementary Table 2.

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

## Acknowledgements
We thank Ryuta Maekawa, Kazuya Matsuo, Angela Yuen, Naoko Miyata, and Kaori Kobayashi for neurite analyses; Yayoi Nozaki for helping the constructions of expression vectors; Ryoko Nakagawa and Kazuo Okanoya for teaching the USV recording techniques; Tomomi Shimogori for teaching the in utero electroporation technique, and Shinichi Hisanaga for PC12D cells. This study is supported by RIKEN BSI funds, MEXT grants (20K06927, 20K21605, 19H03327, 16H04666, 16K07057, 21240031, 25110736), and a grant from the Uehara Memorial Foundation.

## Author contributions
Conceptualization, M.H., K.K., T.Y., and J.A.; Investigation, M.H., K.K., Y.K., H.M., N.T., Y.I., Y.M., A.N., T.Y., and J.A.; Writing—Original Draft, M.H., and J.A.; Writing—Review & Editing, M.H.; Y.K., H.M., A.N., T.Y., and J.A.; Funding acquisition, M.H., J.A.; Supervision, J.A.

## Competing interests
The authors declare no competing interests.
