## [Peer Review File · Communications Biology]

Reviewers' comments:

Reviewer #2 (Remarks to the Author):

Hatayama et al investigate the role of the disease-associated gene *Slitrk1* in brain development. The authors perform an in-depth analysis of behavioral and anatomical parameters in *Slitrk1* KO mice. Differences in ultrasonic vocalizations and noradrenergic fiber density in prefrontal cortex (PFC) are observed. Analysis of the molecular mechanisms involved points to interactions of *Slitrk1* with L1 family proteins and endocytosis. Overall this is an interesting study that reveals the consequences of impaired *Slitrk1* function on the development of noradrenergic circuitry.

Specific comments

1. The manuscript contains a very large amount of data and the results are not always easy to follow. The paper would benefit from reorganization to facilitate reading, for example by reducing supplemental data and grouping data in main figures in line with the text (as an example, now the text moves from Fig. 2 to Fig. 3, then back again to Fig. 2 to discuss metabolite levels and back again to Fig. 3 to discuss VGAT/VGLUT1 staining – this is confusing). The same applies to the order in which figure panels are discussed (for example, Fig. 4D is discussed before 4C in the text). Some supplemental data is not commented on (for example, Fig. S2B, vertical screen test significant difference only at P15). References to supplemental data sometimes appear incomplete (for example, page 7, reference to Supp. FigS4 for increased density of NET-positive fibers likely should include Fig. S5 as well). Finally, the text should be proofread to check for omissions, typos, and to make sure sentences flow well (as an example, 'but females' page 8 probably means 'but less in females' or similar).
2. Images in Fig. 2A and S5A are the same for P7 WT and KO panels. From the text it appears that there should be data on P3 but this is missing.
3. Fig. 4A: differences in mRNA levels for *Sema3A* and *Slits* in WT vs. KO are very small. How was statistical significance determined here? What do data points in graph represent?
4. Page 10, 'Compared with control treatment, *Sema3A* treatment led to decreased and increased complexity of proximal and distal neurites, respectively (Fig. 4C); moreover, it decreased the neurite length (Fig. 4D). However, there was no significant between genotype difference in the complexity (Supplementary Fig. S10-*Sema3a*).' It is unclear to me how the data in 4C, D are related to S10 and what is meant by this sentence.
5. Fig. 4C, D: *Slitrk1*-ECD treatment has no effect on intersections and branch length in *Slitrk1* KO neurons – please discuss.
6. Fig. 6E: the effect of A444S is not significant, yet it is concluded that both S330A and A444S impair inhibitory synapse-inducing ability of *Slitrk1*, this is also repeated in Discussion. This should be corrected. The differences between WT, S330A and A444S are very small, how was significance determined? From the legend it appears that a t-test was used, which is incorrect for multiple conditions here, further it is not clear if data points represent individual cells or independent experiments, in case of the former, corrections should be made to account for independent experiments. In general, statistical analysis should be carefully checked throughout the paper.
7. Fig. 7: this experiment lacks a proper control such as GFP or a non-relevant membrane protein. Currently data is compared to a frame-shifted *Slitrk1* mutant that is predicted to generate a c-terminally truncated ECD fragment. A proper control to account for the experimental procedure (in utero electroporation) should be included.
8. Fig. 8A, proteins co-precipitating with *Slitrk1*: what does the gel in 8A represent? Is this a Coomassie stained gel? I was unable to find the mass spectrometry data that identified L1 family members and dynamin1 as *Slitrk1* interactors, please provide.
9. Page 15, please briefly explain how 'quartz crystal microbalance-based methods' work.
10. Fig. 8B: S330A and A444S should be included for measurements with L1CAM, important for interpretation of endocytosis experiments in 8F-H.
11. Fig. 8F, H: the authors show *Slitrk1* binding affinity for L1 family member Neurofascin is reduced for *Slitrk1* S330A (Fig. 8B). According to the model the authors discuss, *Slitrk1* might displace L1CAM

in a Sema3a-induced endocytosis NRP/L1CAM complex. Based on this model, S330A would be predicted to be less efficient than WT Slitrk1 in suppressing endocytosis, but this is not observed. This apparent discrepancy should be discussed on page 19.

Reviewer #3 (Remarks to the Author):

This manuscript is an extension of authors' previous study on SLITRK1 knockout (KO) mice, which described the altered anxiety-like behavior and abnormalities in noradrenergic functions. They employed both gain-of-function and loss-of-function studies, revealing that Slitrk1 suppresses the noradrenergic projection connectivities that might be involved in a subset of behavioral deficits. Particularly, they asked whether two SLITRK1 missense mutations linked to schizophrenia/bipolar disorder (S330A and A444S) could disrupt any known SLITRK1 functions (e.g. neurite outgrowth). Moreover, they found that L1-CAM binds to SLITRK1 in a nanomolar affinity. I am impressed by large amount of data from various approaches; but the following points should be completely addressed for consideration in Communication Biology.

Major points:

1. Authors need to present summarized/representative results throughout the manuscript by at least three biological replicates. I noted that there are some experiments where the number of samples is below 3. These should be completely addressed.
2. Some of image qualities are not excellent. For example, in Figure 6D, I don't see that authors could conclude any clear conclusions. More importantly, the control experiments appeared not to work – SLITRK1 was reported to induce VGLUT1 clustering in previous studies.
3. Many representative images do not match with the respective quantification results (e.g. Figure 2A and 2B; Figure 3A and Figure 3B; Figure 6A and 6B).
4. I don't understand why the authors treated the SLITRK1 ECD recombinant proteins into cultured LC neurons in Figure 4. I suppose that authors should employ transfection experiments because SLITRK1 is a transmembrane protein. In addition, the representative images do not clearly reflect the message in the Figure 4B.
5. Figure 8: the authors should present the data that show the direct binding of SLITRK1 with Neurofascin and L1CAM, not just by showing Kd values (panel B). Authors did not present any compelling results to demonstrate the interactions of SLITRK1 with these proteins. More rigorous biochemical and biophysical analyses are critical.
6. I am not persuaded by authors' representative images/data that S330A mutations has phenotypes in many cases.
7. Sexual dimorphism is an important/exciting issue in this field. However, authors did not integrate the major findings related to this issue in the current manuscript. Could authors show for example that differences in the norepinephrine levels between sexes are causally involved in behavioral deficits and neurite outgrowth/synaptic impairments?
8. Data presentations: this is serious – bar graphs should be presented in a consistent manner throughout the paper.

Minor points:

1. Discussion section is too lengthy. Authors need to remove/trim considerable parts and move them to the Introduction section.
2. Authors need to discuss in detail why varicosity size in the NET fibers are opposite in SLITRK1 KO mice during development (i.e., P7 vs. 6M).
3. Page 8, line 4: "but females" should be changed to "but not females"
4. Figure S4: These data should be quantified.
5. Where are the representative images for Figure 3I and 3J in Figure 3H?
6. Where are the representative images for BSA groups in Figure 8F?
7. Statistics: # should be added in the legend of Figure 2B. In addition, ++ need to removed. In Figure

2E, † need to be corrected. Importantly, authors should be very careful to indicate all the statistics in data and legend (e.g. statistics missed in Figure 2E for MHPG and MHPG/NA groups).
8. Figure 2B: further experiments are necessary for female mice.

Reviewer #4 (Remarks to the Author):

Summary:

This manuscript reports an array of neurodevelopmental differences in mice lacking SLITRK1, which is a transmembrane protein associated with OCD-related disorders and known to function in neurite outgrowth and synapse formation. The paper builds on the authors' previously published work, which implicated noradrenergic mechanisms in the anxiety-like behaviors of *slitrk1*-deficient mice. Here, the authors report structural, behavioral, and neurochemical differences in *slitrk1*-deficient mice, with an emphasis on the neonatal prefrontal cortex. The authors also report two novel missense SLITRK1 mutations associated with schizophrenia and bipolar disorder and link these mutations to structural and functional deficits in the noradrenergic system. The paper is ambitious in scope, technically rigorous, and provides further evidence that cellular and molecular differences in noradrenergic signaling may contribute to numerous neurodevelopmental disorders, including OCD, schizophrenia, and bipolar disorder.

Major points:

This work is of interest to at least three specialized audiences - those who examine SLITRK protein functions in neurodevelopment, those interested in the development of the prefrontal cortex, and those interested in the cellular and molecular changes that contribute to OCD and other pervasive but poorly understood neurodevelopmental disorders.

With 9 Figures and 13 Supplementary Figures, the paper is ambitious in a way that ultimately compromises its cohesion and readability. The authors characterize a knock-out mouse at the behavioral, anatomical, and molecular levels, provide evidence for changes to multiple neurotransmitter pathways, perform a genome wide association study, report novel SLITRK1-interacting proteins, and examine the complex molecular mechanisms mediated by these protein interactions. The reason why some figures are relegated to Supplementary files (while others are not) is somewhat difficult to discern. It is an impressive and technically rigorous body of data, but it may benefit from being divided into two smaller, more cohesive manuscripts.

Two general issues regarding the clarity of the manuscript should be addressed. First, the final paragraph of the introduction should be revised and expanded to articulate the overall rationale and major findings of the paper with greater specificity. Second, the Results section could benefit from clear statements of rationale as each experiment is introduced and described. This was done nicely in the Discussion, but bears repeating in the Results as well.

In Figures 2 and 3 (and others), mean values of WT controls are set to 1, but no mention of the absolute values can be found in the figure legends or results text. A supplementary data file containing absolute measurements would be a positive addition to the supplementary materials.

Figure 5, which describes the genome wide association study that identified new SLITRK1 missense mutations, may be better suited to the Supplementary data.

The claim that *Slitrk1*-knockout mice exhibit reduced GABAergic synapse density in the PFC is not adequately supported by the evidence presented in Figure S8. One or more additional inhibitory synapse markers would strengthen this claim considerably.

Minor Points:

In Figures 2-4, the WK labels below the X axes are redundant with the shading of the bars and the associated key. Removing the WK labels would improve the clarity of these figures.

Statistical analyses are appropriate throughout, and the level of experimental detail provided in the Methods section is adequate for reproducibility.

Point-by-point responses to the reviewers' comments.

RE: Hatayama et al. "SLITRK1-mediated noradrenergic projection suppression in the neonatal prefrontal cortex." (COMMSBIO-21-2501A)

Major points of revision:

1. We reorganized the manuscript for enhanced readability. This was done by describing the overall rationale and major findings in the Introduction section (reviewer #4), correcting the arrangement and the order of the results (reviewer #2), clarifying the logical flow in the Results section (reviewer #4), trimming the lengthy discussion (reviewer #3), and by modifying the graph design to be consistent (reviewer #3). The number of figures was increased from 9 to 11. The number of Supplementary figures was reduced from 12 to 9.
2. Quality of the data was improved. This included removing the results with low experimental powers (Figure 3H-J, S8, S9D, E), increasing "n" (Figure 2b), replacing the representative images (Figures 2a, 3a, and d, new Figure 6b, new Supplementary Figure 7), indicating original data (new Supplementary Table 2), and adding a proper control (new Figure 9) or new results (new Figure 10b, new Figure S8b-d).
3. The Methods section was supplemented with some essential information to increase the reproducibility. Methods for the experiments in the Supplementary information were included in the supplementary text.

Point-by-point responses to reviewers' comments:

Reviewer #2:

Hatayama et al investigate the role of the disease-associated gene *Slitrk1* in brain development. The authors perform an in-depth analysis of behavioral and anatomical parameters in *Slitrk1* KO mice. Differences in ultrasonic vocalizations and noradrenergic fiber density in prefrontal cortex (PFC) are observed. Analysis of the molecular mechanisms involved points to interactions of *Slitrk1* with L1 family proteins and endocytosis. Overall this is an interesting study that reveals the consequences of impaired *Slitrk1* function on the development of noradrenergic circuitry.

Specific comments

1. The manuscript contains a very large amount of data and the results are not always easy to follow. The paper would benefit from reorganization to facilitate reading, for example by reducing

supplemental data and grouping data in main figures in line with the text (as an example, now the text moves from Fig. 2 to Fig. 3, then back again to Fig. 2 to discuss metabolite levels and back again to Fig. 3 to discuss VGAT/VGLUT1 staining – this is confusing).

The same applies to the order in which figure panels are discussed (for example, Fig. 4D is discussed before 4C in the text). Some supplemental data is not commented on (for example, Fig. S2B, vertical screen test significant difference only at P15).

Response: After reading the three reviewers' comments, we realized that the previous manuscript was not well organized and forced the readers to seek the data many times. According to this comment, we have reorganized the results so that the readers can follow the study without unnecessarily going back-and-forth. The Supplementary Figures were limited to only those figures which the readers could follow the study without seeing, and the number was reduced from 12 to 9.

References to supplemental data sometimes appear incomplete (for example, page 7, reference to Supp. FigS4 for increased density of NET-positive fibers likely should include Fig. S5 as well). Finally, the text should be proofread to check for omissions, typos, and to make sure sentences flow well (as an example, 'but females' page 8 probably means 'but less in females' or similar).

Response: We have corrected and checked the reference to the supplemental data. Several instances, including the one pointed out, were corrected to improve the flow. The revised text was finally corrected by a professional scientific Editor once again.

2. Images in Fig. 2A and S5A are the same for P7 WT and KO panels. From the text it appears that there should be data on P3 but this is missing.

Response: Figure 2A, B and previous S5A, B, D were merged in a new Figure 2a-c. Figure 2C, D was merged with previous S5E to a new Figure 2d, e. P3 images were added to Figure 2a, d. In the revised Figure 2, readers can now see the data for NET staining at P3, P7, 5W, and 6M at a glance.

3. Fig. 4A: differences in mRNA levels for Sema3A and Slits in WT vs. KO are very small. How was statistical significance determined here? What do data points in graph represent?

Response: Statistical tests were performed by two-tailed unpaired t-test and two-way ANOVA (genotype and sex as main factors). A data point indicates a mean value for each mouse. The mean values were derived from three qPCR experiments that were performed in duplicate. In this

revision, we carried out multiple test correction as per the Benjamini–Hochberg procedure and confirmed that the changes in the three genes were significant after the multiple test correction. These points have been described in the Figure legend of new Figure 6.

4. Page 10, ‘Compared with control treatment, Sema3A treatment led to decreased and increased complexity of proximal and distal neurites, respectively (Fig. 4C); moreover, it decreased the neurite length (Fig. 4D). However, there was no significant between genotype difference in the complexity (Supplementary Fig. S10-Sema3a).’ It is unclear to me how the data in 4C, D are related to S10 and what is meant by this sentence.

Response: We apologize for the confusion caused by our unclear description. “Control treatment” indicated to the BSA-treated one, referring to the BSA-Sema3A difference in the KO samples. To avoid confusion, we have moved the graphs for between-genotype difference (previous Figure S10) to the bottom of new Figure 6c and referred to the individual graph in the main text.

5. Fig. 4C, D: Slitrk1-ECD treatment has no effect on intersections and branch length in Slitrk1 KO neurons – please discuss.

Response: We agree that this is an important finding. In the last paragraph of *Molecular mechanism underlying Slitrk1-mediated suppression of NA projections* chapter, we have discussed this as follows: “The fact that the neurite suppressive effect of SLITRK1 ECD requires Slitrk1 in LC neuron suggests that Slitrk1 and Slitrk1 ECD may compete for the L1 family proteins. However, we cannot exclude other possibilities, such as homophilic interaction via Slitrk1 ECDs or its binding to unidentified targets.”

6. Fig. 6E: the effect of A444S is not significant, yet it is concluded that both S330A and A444S impair inhibitory synapse-inducing ability of Slitrk1, this is also repeated in Discussion. This should be corrected.

Response: This comment may have been raised due to an unclear indication of statistical significance in Figure 6E graph. In the bar graphs of the revised manuscript, we have consistently used rotated-square-brackets to indicate the two groups for the comparison. The corresponding figure was indicated in new Supplementary Figure 7.

The differences between WT, S330A and A444S are very small, how was significance determined? From the legend it appears that a t-test was used, which is incorrect for multiple conditions here, further it is not clear if data points represent individual cells or independent

experiments, in case of the former, corrections should be made to account for independent experiments. In general, statistical analysis should be carefully checked throughout the paper.

Response: In the revised manuscript, we carried out Dunnett's or Steel's test for the mutant analysis (one-to-many comparison). This has also been written in *Experimental design and statistical analysis* subsection of the *Methods* section and in the Figure legend.

7. Fig. 7: this experiment lacks a proper control such as GFP or a non-relevant membrane protein. Currently data is compared to a frame-shifted Slitrk1 mutant that is predicted to generate a c-terminally truncated ECD fragment. A proper control to account for the experimental procedure (in utero electroporation) should be included.

Response: Accordingly, new *in utero* electroporation experiments for GFP and WT were done and the results are presented in new Figure 9. Results of the statistical tests were also indicated for the new control group.

8. Fig. 8A, proteins co-precipitating with Slitrk1: what does the gel in 8A represent? Is this a Coomassie stained gel? I was unable to find the mass spectrometry data that identified L1 family members and dynamin1 as Slitrk1 interactors, please provide.

Response: The gel in Figure 8A (new Figure 10a) represents the Coomassie stained gel. We have explained it in the figure legend. Original mass spectrometry data were indicated in the new Supplementary Table 2.

9. Page 15, please briefly explain how 'quartz crystal microbalance-based methods' work.

Response: We have explained this in the Results section text as follows: "Regarding L1 family proteins, the physical interactions between L1 family-ECDs and SLITRK1 ECD-immunoglobulin Fc domain-fusion protein (SLITRK1-WT-Fc) were quantitatively assessed using a quartz crystal microbalance that measures mass variation per unit area by measuring the change in the frequency of a quartz crystal resonator. The addition of aliquots of L1 family-ECDs solution to SLITRK1 ECD-Fc immobilized on a thin plate attached to a crystal resonator showed a concentration-dependent decrease in the frequency, enabling us to determine the kinetic parameters."

10. Fig. 8B: S330A and A444S should be included for measurements with L1CAM, important for interpretation of endocytosis experiments in 8F-H.

Response: We have included the results of the QCM analysis for L1CAM, SLITRK1 S330A, and A444S in new Figure 10b. The results indicate that the L1CAM-Slitrk1 interaction kinetics was not affected by S330A or A444S. Based on these result, relevant text in the end of the *Results* section has been modified as follows: "S330A and A444S reduced the levels of Sema3a+FM4-

64+Nrp1+L1CAM+ vesicles comparable to those of WT (Figure 10f, h) in agreement with the comparable L1CAM-binding properties between WT and the two variants.”

11. Fig. 8F, H: the authors show Slitrk1 binding affinity for L1 family member Neurofascin is reduced for Slitrk1 S330A (Fig. 8B). According to the model the authors discuss, Slitrk1 might displace L1CAM in a Sema3a-induced endocytosis NRP/L1CAM complex. Based on this model, S330A would be predicted to be less efficient than WT Slitrk1 in suppressing endocytosis, but this is not observed. This apparent discrepancy should be discussed on page 19.

Response: Because the binding to L1 was comparable between SLITRK1 WT and S330A as shown above, the endocytosis-suppressing activities of the two proteins may be equivalent.

Reviewer #3:

This manuscript is an extension of authors' previous study on SLITRK1 knockout (KO) mice, which described the altered anxiety-like behavior and abnormalities in noradrenergic functions. They employed both gain-of-function and loss-of-function studies, revealing that Slitrk1 suppresses the noradrenergic projection connectivities that might be involved in a subset of behavioral deficits. Particularly, they asked whether two SLITRK1 missense mutations linked to schizophrenia/bipolar disorder (S330A and A444S) could disrupt any known SLITRK1 functions (e.g. neurite outgrowth). Moreover, they found that L1-CAM binds to SLITRK1 in a nanomolar affinity. I am impressed by large amount of data from various approaches; but the following points should be completely addressed for consideration in Communication Biology.

Major points:

1. Authors need to present summarized/representative results throughout the manuscript by at least three biological replicates. I noted that there are some experiments where the number of samples is below 3. These should be completely addressed.

Response: We noticed that the previous Supplementary Figure 8 contained the results of $n = 2$ and $n = 3$. The figure and related description were removed from this study. In relation to this revision, we removed the quantitative analyses of synapses in Slitrk1 KO mice (previous Figure 3H-J, S9D, E). This was done because we had to simplify the entire study in response to the comments from the other reviewers. The synaptic phenotype will be described elsewhere.

2. Some of image qualities are not excellent. For example, in Figure 6D, I don't see that authors could conclude any clear conclusions. More importantly, the control experiments appeared not to work – SLITRK1 was reported to induce VGLUT1 clustering in previous studies.

Response: We replaced Figure 6D (new Supplementary Figure 7) images. In our experiments, SLITRK2 induced both inhibitory and excitatory synapses (new Supplementary Figure 7). In this regard, a control may be working. Concerning the discrepancy with previous reports, several experimental parameters are different among the studies (see the Table below). Most critically, previous studies used artificial signal peptide sequence and N-terminal epitope tag, and the timing of the co-culture was earlier than ours. The difference between SLITRK1 and SLITRK2 in our condition may reflect differences in some inherent properties of these proteins. We feel that it would be worth reporting the result in this condition.

	species of cDNA	tag	vector	promotor	host cell	source of neurons	co-culture	harvest
this study	human	(-)	pCAGGS	CAG	293T	E18 rat hippocampus	DIV15	DIV17
Takahashi et al., 2012	mouse	HA	spYFP-C1	CMV	COS7	E18 rat hippocampus	DIV8-9	DIV9-10
Kang et al., 2016	mouse?	HA	pDisplay	CMV	293T	E18 rat hippocampus	DIV9	DIV12

3. Many representative images do not match with the respective quantification results (e.g. Figure 2A and 2B; Figure 3A and Figure 3B; Figure 6A and 6B).

Response: We agree with this comment. We have replaced the images in Figure 2A (new Figure 2a), Figure 3A (new Figure 3a), Figure 4B (new Figure 6b), Figure 6A (new Figure 8b), Figure 6D (new Figure S7), and Figure 8F (new Figure 10f). In addition, varicosity images were indicated as insets in some Figure 2 and Figure 3 images.

4. I don't understand why the authors treated the SLITRK1 ECD recombinant proteins into cultured LC neurons in Figure 4. I suppose that authors should employ transfection experiments because SLITRK1 is a transmembrane protein. In addition, the representative images do not clearly reflect the message in the Figure 4B.

Response: SLITRK1 ECD is known to be secreted after cleavage by α/γ secretase (ref 17. Kajiwara et. al., 2009) as described in the *Introduction* section. In the revised manuscript, we have added the following text in the *Results* section before describing the results with SLITRK1 ECD: “We also investigated the effect of SLITRK1 ECD because SLITRK1 ECD is known to be cleaved by α/γ secretase at the transmembrane region ¹⁷”. To consider the role of Slitrk1 in LC neurons (cell-autonomous function), we compared the LC neurons from Slitrk1 WT and KO mice. In the revised figure (new Figure 6b, c), we replaced the representative images in Figure 4B and merged graphs in Figure 4C and Figure S10. We hope that these corrections can improve the readability and visibility.

5. Figure 8: the authors should present the data that show the direct binding of SLITRK1 with Neurofascin and L1CAM, not just by showing Kd values (panel B). Authors did not present any compelling results to demonstrate the interactions of SLITRK1 with these proteins. More rigorous biochemical and biophysical analyses are critical.

Response: Accordingly, we performed pull-down assay using purified SLITRK1 ECD, Neurofascin ECD, and L1CAM ECD proteins. The results are indicated in new Supplementary Figure 8c. The results for the pulldown assay using brain lysate and SLITRK1 ECD are also indicated for L1CAM, Neurofascin, and NCAM (new Supplementary Figure 8b). In addition, we measured the Kd values the SLITRK1 mutants (new Figure 10b), and the original result for mass spectrometry was indicated in new Supplementary Table 2 upon suggestion from another reviewer.

6. I am not persuaded by authors' representative images/data that S330A mutations has phenotypes in many cases.

Response: Representative images for Figure 6D (new Supplementary Figure 7) were replaced with better images. The figures indicating S330A's effect on "later" neurite development (previous Figure S10I-K) were incorporated into new Figure 8 (e-g). These figures most clearly indicate the S330A phenotypes. We agree with the reviewer in that S330A phenotypes are not so drastic. However, this may be a very important mutation with not only pathological, but also evolutionary significance (please see the last paragraph of the *Discussion* section). We therefore conducted the experiments carefully. In this regard, we added the following description in the *Experimental design and statistical analysis* subsection of the *Methods* section: "The experiments for assessing the new SLITRK1 mutations were carried out in a plasmid identity-blinded manner. The expression constructs for SLITRK1 WT and its mutants were verified by sequencing after each plasmid preparation."

7. Sexual dimorphism is an important/exciting issue in this field. However, authors did not integrate the major findings related to this issue in the current manuscript. Could authors show for example that differences in the norepinephrine levels between sexes are causally involved in behavioral deficits and neurite outgrowth/synaptic impairments?

Response: We agree with the reviewer. We observed several sexual dimorphisms in *Slitrk1* KO phenotypes (body weight, USV, NA contents in PFC, and SERT varicosity size in PFC) and discussed the possible basis of this observation (e.g. VMAT2, COMT) in the first subsection of the *Discussion*. The possible causal relationships were described for the synaptic, behavioral, and developmental phenotypes in the same subsection. In addition, we referred to the male-

predominant occurrence of early-onset OCD in the “*Excessive NA projections as a disease core mechanism underlying OCD*” subsection.

8. Data presentations: this is serious – bar graphs should be presented in a consistent manner throughout the paper.

Response: Accordingly, we changed the graph design for consistency. All data for WT mice are indicated by open circles or bars and those for KO mice are indicated by red circles and bars. Data for SLITRK1 proteins and its mutants are indicated by different colors (WT, blue; S330A, green; A444S, yellow). Statistical significance was indicated in a consistent manner.

Minor points:

1. Discussion section is too lengthy. Authors need to remove/trim considerable parts and move them to the Introduction section.

Response: We moved the background information on the neural circuit basis of OCD and molecular determinants of the monoaminergic fibers from the *Discussion* to *Introduction*. This revision may be beneficial for the readers to read the *Results* more easily.

2. Authors need to discuss in detail why varicosity size in the NET fibers are opposite in SLITRK1 KO mice during development (i.e., P7 vs. 6M).

Response: We added the following discussion in the first chapter of the Discussion section. “Contrarily, the NET⁺ varicosity size in the PFC of male Slitrk1 KO was larger at P7 but smaller than those of WT at 5W or 6M (Figure 2). While the increase at P7 can be interpreted as a feedback from excessive NA via $\alpha 2$ autoreceptor, the decrease at later stages suggests the presence of some adaptive mechanisms for the excessive NA. As a candidate mediator of such adaptive response, VMAT2 should be noted because VMAT2 is a critical regulator of presynaptic NA storage in brain, and VMAT2 expression is dynamically regulated both during development and upon acute and chronic drug exposure (Eiden and Weihe, 2011).”

3. Page 8, line 4: “but females” should be changed to “but not females”

Response: We have corrected it.

4. Figure S4: These data should be quantified.

Response: We have quantified the NET⁺ fiber area. The results are indicated at the bottom of the images.

5. Where are the representative images for Figure 3I and 3J in Figure 3H?

Response: We have removed the corresponding figures in this revision. Please see the response to Major point 1.

6. Where are the representative images for BSA groups in Figure 8F?

Response: The representative images for the BSA control experiments were added to Figure 8F (new Figure 10f).

7. Statistics: # should be added in the legend of Figure 2B. In addition, †† need to be removed. In Figure 2E, † need to be corrected. Importantly, authors should be very careful to indicate all the statistics in data and legend (e.g. statistics missed in Figure 2E for MHPG and MHPG/NA groups).

Response: Symbols for the statistical test results were changed to be consistent throughout the graphs. Complete information for the statistical tests was included in the figure legends.

8. Figure 2B: further experiments are necessary for female mice.

Response: An additional experiment was done for female mice. Total n became eight for each genotype.

Reviewer #4 (Remarks to the Author):

Summary:

This manuscript reports an array of neurodevelopmental differences in mice lacking SLITRK1, which is a transmembrane protein associated with OCD-related disorders and known to function in neurite outgrowth and synapse formation. The paper builds on the authors' previously published work, which implicated noradrenergic mechanisms in the anxiety-like behaviors of *slitrk1*-deficient mice. Here, the authors report structural, behavioral, and neurochemical differences in *slitrk1*-deficient mice, with an emphasis on the neonatal prefrontal cortex. The authors also report two novel missense SLITRK1 mutations associated with schizophrenia and bipolar disorder and link these mutations to structural and functional deficits in the noradrenergic system. The paper is ambitious in scope, technically rigorous, and provides further evidence that cellular and molecular differences in noradrenergic signaling may contribute to numerous neurodevelopmental disorders, including OCD, schizophrenia, and bipolar disorder.

Major points:

This work is of interest to at least three specialized audiences - those who examine SLITRK protein functions in neurodevelopment, those interested in the development of the prefrontal cortex, and those interested in the cellular and molecular changes that contribute to OCD and other pervasive but poorly understood neurodevelopmental disorders.

With 9 Figures and 13 Supplementary Figures, the paper is ambitious in a way that ultimately compromises its cohesion and readability. The authors characterize a knock-out mouse at the behavioral, anatomical, and molecular levels, provide evidence for changes to multiple neurotransmitter pathways, perform a genome wide association study, report novel SLITRK1-interacting proteins, and examine the complex molecular mechanisms mediated by these protein interactions.

The reason why some figures are relegated to Supplementary files (while others are not) is somewhat difficult to discern. It is an impressive and technically rigorous body of data, but it may benefit from being divided into two smaller, more cohesive manuscripts.

Response: We thank you the thoughtful comments. Together with the other reviewers' comments, we have reduced the number of supplementary figures and removed the synaptic phenotype data that did not strongly correlate with our study's aim. We selected the main figures keeping in mind that most readers could understand the paper without referring to the Supplementary figures and tables.

Two general issues regarding the clarity of the manuscript should be addressed. First, the final paragraph of the introduction should be revised and expanded to articulate the overall rationale and major findings of the paper with greater specificity. Second, the Results section could benefit from clear statements of rationale as each experiment is introduced and described. This was done nicely in the Discussion, but bears repeating in the Results as well.

Response: We revised the final paragraph as follows: "In this study, we firstly examined the neonatal phenotypes of Slitrk1-deficient mice. Because abnormalities in NA fiber development were observed, we investigated the molecular mechanism underlying Slitrk1-mediated control of the neurite development. Further, we conducted a re-sequencing analysis of patients with schizophrenia (SCZ) and bipolar disorder (BPD) to identify functionally defective and significantly enriched missense mutations. The analysis identified a SLITRK1 mutation that affects the NA fiber development-controlling and L1 family protein-binding abilities of SLITRK1. Finally, we sought to discuss the pathogenesis of OCD, focusing on the role of neonatal NA-mediated neural circuit modification."

We have added text that clarifies the logical flow in several subsections of the *Result* section.

In Figures 2 and 3 (and others), mean values of WT controls are set to 1, but no mention of the absolute values can be found in the figure legends or results text. A supplementary data file containing absolute measurements would be a positive addition to the supplementary materials.

Response: We understand the comments. Concerning the quantitative immunostaining results (Figure 2, Figure 3, new Figure 9) or qPCR results (new Figure 6a), most data consisted of results from several experiments. Because the experiment-to-experiment variations of the absolute values were rather strong, normalization (e.g. WT mean value = 1) was essential. We consider that indication of the absolute values for the varicosity sizes may be meaningful. The absolute values were included in the figure legends of Figure 2c, 2e, 3c, 3f, 3i, and 3l. Concerning the monoamine quantification, we have included the values in Supplementary Figure 5.

Figure 5, which describes the genome wide association study that identified new *SLITRK1* missense mutations, may be better suited to the Supplementary data.

Response: We think this result is very important in terms of clinical genetics. Although the sample size is not large, it can be utilized for meta-analysis in future. In addition, it contains a meaningful result in terms of evolutionary biology. In these regards, the result should be presented as a main figure.

The claim that *Slitrk1*-knockout mice exhibit reduced GABAergic synapse density in the PFC is not adequately supported by the evidence presented in Figure S8. One or more additional inhibitory synapse markers would strengthen this claim considerably.

Response: We removed all results concerning the quantitative analysis of the synapse markers in *Slitrk1* KO mice. Please see our first response.

Minor Points:

In Figures 2-4, the WK labels below the X axes are redundant with the shading of the bars and the associated key. Removing the WK labels would improve the clarity of these figures.

Response: We removed the WK label, and the results in all the graphs are indicated in a consistent color code (WT = white, KO = red).

Statistical analyses are appropriate throughout, and the level of experimental detail provided in the Methods section is adequate for reproducibility.

Response: We added the experimental design for the SLITRK1 mutant analysis in the *Methods* section.

REVIEWERS' COMMENTS:

Reviewer #2 (Remarks to the Author):

The efforts that the authors have made to change the flow of the manuscript figures and text have resulted in a much improved manuscript. The authors have convincingly addressed my comments and made necessary clarifications. The manuscript is ready for publication in my opinion.

Reviewer #3 (Remarks to the Author):

The authors have addressed most of my comments satisfactorily, except one: Pulldown experiments do not convincingly address the direct protein interactions. I strongly urge the authors to show the direct binding assays using recombinant proteins encoding Slitrk1 and putative ligands identified from the current study. Otherwise, the authors may want to remove a whole set of data that claim the interactions.

Reviewer #4 (Remarks to the Author):

The authors have adequately addressed the suggested revisions. Their findings enhance our understanding of schizophrenia/bipolar-related gene functions at the cellular and sub-cellular level.